

# Assessing water supply capacity in a complex river basin under climate change using the logistic eco-engineering decision scaling framework

Daeha Kim[1], Jong Ahn Chun[1], Si-Jung Choi[2]

[1]APEC Climate Center, Busan, 48058, South Korea
[2]Korea Institute of Civil Engineering and Building Technology, Gyeonggi-do, 10223, South Korea

*Correspondence to*: Si-Jung Choi (sjchoi@kict.re.kr)

**Abstract.** Climate change is a global stressor that can undermine water management policies developed under the assumption of stationary climate, necessitating robust solutions to reducing the risk of system failures for uncertain future

climates. While the response-surface-based assessments have provided convenience to explore responsive behaviours of expected system performance to climatic stresses, they were unable to predict the risk of system failures from individual climate projections. In this study, we proposed to use the logistic regressions for evaluating the probability of non-successive outcomes against pre-defined thresholds directly from climate projections, which may be more informative for decision making processes than the expected performances. As a case study, water supply and ecological reliabilities within a large

river basin were assessed by combining the eco-engineering decision scaling framework and the logistic regressions. The impact assessment for the South Korean river basin showed that optimal water supply performance at the sub-basins were expected to be satisfactory for the upcoming 20 years of 2020-2039, while the human-demand-only operations could lower the ecological reliabilities. When considering ecological demands in water operations to reduce the ecological vulnerabilities, the stakeholders should accept increasing risks of unsatisfactory supply at the sub-basins with low demands. This study

highlights that binary conversions of the performance metrics from the stress tests allow users to measure the risks of system failures varying across sub-components and standpoints with minimal computational costs.

## 1 Introduction

Climate change is a global stressor that poses prodigious challenges to long-term management of water resources. While water infrastructures have been constructed across the globe to sustain human livelihood and activities, those assets

have been traditionally managed by heuristic operation policies developed under the assumption of stationary climate (Cosgrove and Loucks, 2015; Cully et al., 2016). The probabilistic behaviours of hydro-climatological variables, however, can be significantly altered by the warming atmosphere; thereby, the traditional management is expected to become increasingly vulnerable (Brown et al., 2015; Georgakakos et al., 2012). The adaptive operations may reduce risks from




climatic change and variability (e.g., Xu et al., 2015; Eum and Simonovic, 2010), but are often difficult to be utilized due to deeply uncertain climate and socioeconomic conditions (Weaver et al., 2013; Steinschneider and Brown, 2012).

For assessing impacts of climate change on water resources systems, the primary approach was to investigate outputs of relevant system models forced by projections of the general circulation models (GCMs) under hypothetical greenhouse gas (GHG) emission scenarios (Brown et al., 2012; Wilby and Dessai, 2010). Though it was possible to derive potential adaptive solutions from the given climate projections (e.g., Vano et al., 2010), the scenario-led (or namely "top-down") approaches resulted in underutilization of the achieved solutions in practice (Weaver et al., 2013; Brown and Wilby, 2012). This type of assessments takes the 'predict-then-act' paradigm for which the first prerequisite is accurate and reliable predictions. However, substantial efforts are still necessary to deliver improvement in climatic predictions (Shapiro et al., 2010; Piao et al., 2010; Barron, 2009; Goddard et al., 2009; Doherty et al., 2009). This implies that previous scenario-led assessments were conducted before the first prerequisite was sufficiently satisfied. The GCM projections are often biased by inappropriate model formulations and/or imperfectly understood physical processes (Stevens and Bony, 2013; Deser et al., 2012; Dufresne and Bony, 2008; Stainforth et al., 2005); thus, they may be subject to unacceptable uncertainty and potential regrets. Given the high risk costs contained in GCM-led analyses, adaptive solutions from a small collection of GCM projections might be discarded by decision-makers' risk-aversion behaviours (Brown et al., 2012).

To overcome the weakness of the scenario-led strategies in practical decision support, alternative approaches within the 'robust decision' paradigm have emerged (e.g., Hadka et al., 2015; Whateley et al., 2014; Lampert and Groves, 2010). These decision-centric (or namely "bottom-up") approaches seek robust solutions that minimize adverse effects of climatic stresses on systems of interest. Whereas the top-down framework focuses on the most likely future conditions to maximize expected utility, the bottom-up approaches pay attention to sensitivity (or vulnerability) of system performance to climatic stressors (Weaver et al., 2013; Brown et al., 2012). Examples include the decision scaling (Brown et al., 2012), the dynamic adaptive policy pathways (Haasnoot et al., 2013), the real option analysis (Woodward et al., 2014), the Info-gap decision theory (Korteling et al., 2013), and the robust decision making (Lempert and Groves, 2010) among others. The bottom-up approaches assess system performance under diverse climatic exposures (i.e., hypothetical stress tests) first. Then, the GCM projections are simply used as indicators of system vulnerability in the future (i.e., impact assessment). This risk-based framework accepts the irreducible uncertainty in climate predictions as an inevitable part of long-term planning, and guides decision-makers toward low-regret strategies for sustainable system performance under non-stationary climate (Poff et al., 2016).

Among the bottom-up approaches, the response-surface-based methods have provided convenience to visualize system performance within a full range of plausible climatic exposures. By developing the "response surfaces" that directly relate performance metrics to climatic changes via the stress tests (e.g., Prudhomme et al., 2010; Brown et al., 2012; Whatley et al., 2014; Steinschneider et al., 2015a), users may thoroughly explore system sensitivity to climatic stressors beyond GCM projections. While the early works were intended to investigate changes of performance indicators in climatic domains (e.g., Turner et al., 2014), this approach have evolved to account for system behaviours varying across sub-components and



adaptation options (Schlef et al., 2017; Steinschneider et al., 2015), to define multi-objective climatic boundaries (Cully et al., 2016), and to find robust solutions between contrasting interests (Poff et al., 2016). These efforts enabled users to consider various facets of system performances in the bottom-up impact assessments.

However, all the assessments using the response surfaces have focused on the "expected" performance rather than the risk of system failures. The system performances predicted only by climatic changes are likely to contain considerable uncertainty because of other factors contributing to variation of the selected performance indicators. Whateley and Brown (2016) showed that variations of water supply performance in hydrologic systems were significantly attributed to short-term climate variability over planning horizons. Indeed, the simplifications and assumptions used in the stress tests may introduce appreciable biases in the response surfaces (Kay et al., 2014; Steinschneider et al., 2015b), implying that even GCMs indicating satisfactory performances in the response surfaces are subject to the risk of system failures. In other words, the risk of system failures could be ill-defined if one counts GCMs on the unsatisfactory zone of the response surfaces.

With pre-defined performance thresholds, the probabilities of system failure against the thresholds may be more necessary information than the expected performance in decision-making processes due to the uncertainty of the response surfaces. Particularly when contrasting performance criteria are assessed by the stress tests, the risk of system failures may become essential information. Poff et al. (2016) proposed the eco-engineering decision scaling framework that overlaps multiple response surfaces to identify the climate zone mutually satisfying engineering and ecological performance criteria. The size of the mutual zone was used as a proxy indicator of system robustness to climatic stressors, leading users to adaptation options balanced between conflicting interests. However, in this innovative approach, GCM projections were not superimposed on the response surfaces. In part, this may be because the GCM counts may not be meaningful information when the mutual zone is too narrow. In this case, it would be better to predict the risks of failures directly from individual GCMs instead of the expected performances; however, it may require expensive efforts when a common uncertainty analysis is employed (e.g., Kay et al., 2014; Steinschneider et al., 2015b).

In this work, therefore, we proposed to modify the eco-engineering decision scaling framework by combining it with the logistic regression analysis. The modified framework allows users to predict the probability of system failure (or success) against pre-defined thresholds by simple binary conversions of performance metrics gained from the stress tests. As a case study, we attempted to assess water supply capacities in a large river basin regulated by multiple dams and reservoirs under diverse climate exposures. The case study shows how the risks of engineering and ecological failures change across sub-components of the complex river basin according to operational strategies. For the bottom-up assessment, we embedded a sequential optimization scheme in stochastic stress tests to quantify water supply reliabilities. The water supply and ecological risks were graphically showed by drawing the boundaries indicating 95% probability of successive performance for every sub-component together. To evaluate future risks of system failures within the eco-engineering framework, GCM projections were superimposed on the climatic space.



## 2 Description of the study site and data

### 2.1 Geum River Basin

The study site is the Geum River Basin located in the west-central part of South Korea with a total area of 9,915 km$^2$ (Figure 1). The mean and the highest elevations in the river basin are 85 m and 1,596 m above the sea level, respectively. The mean

basin slope is 16.7 % from the elevation profile. The total length of the main channel is approximately 402 km. The river basin has a semi-humid climate with monsoonal summer seasons. Wet air masses moving from the North Pacific make hot and humid summer seasons, whereas winter seasons are dry and cold due to the Siberian high pressure. Approximately, 60-70% of annual precipitation falls in the monsoonal summer seasons (June to September) (KMA, 2011). Streamflow in the river basin usually peaks in the middle of monsoon seasons by tropical or extra-tropical cyclones, while occasional

precipitation events limitedly recharge the river basin during dry seasons (October to May). Snowmelt minimally influences streamflow variations due to small winter precipitation (Bae et al., 2008).

The Geum River Basin is officially divided into 14 sub-basins for administrative purposes along the geomorphological boundaries. The sub-basin areas vary between 120 and 1,856 km$^2$ with an average of 708 km$^2$. 60% of the river basin is covered by forests, while the agricultural areas account for 18%. The forest covers and agricultural lands in the 14 sub-basins

occupy 33-83% and 4.6-42% of the sub-basin areas, respectively. The sub-basins with relatively large agricultural lands tend to have small forest covers. The urban areas are 5.3% of the river basin in total. According to the Korea Forest Service (http://forest.go.kr), the soils across the Geum River Basin generally have moderate to high infiltration capacity, implying that the sub-surface processes significantly contribute to event flow generations.

Human interventions affect the flow regimes in the Geum River Basin. The main channel is regulated by two large dams

serially connected for water supplies and flood controls. The Yongdam Dam located in the upper river basin has an effective storage capacity of 809 Mm$^3$, while the Daecheong Dam at the middle of the main channel has a larger capacity of 1,040 Mm$^3$. The water storages in both dams are delivered to several sub-basins through water distribution systems developed for municipal and industrial (M&I) water demands, making non-geomorphological human-made connectivity between the sub-basins. The two large dams supply water to the demand sectors in outsides of the river basin through the distribution systems;

hence, inter-basin water transfers may conflict with water demands within the river basin. During monsoon seasons, Yongdam and Dacheong Dams should reduce their storage limits by 137 Mm$^3$ and 250 Mm$^3$ for flood control, respectively. Additionally, many small-size local reservoirs are widespread across the river basin to sustain irrigated agriculture (mostly for planting paddy rice). Though 95% of the small reservoirs have minimal storage capacities below 1 Mm$^3$, their gross capacity is more than 320 Mm$^3$ and thus significantly alters natural flow regimes. The total storage capacities of the

agricultural reservoirs in the 14 sub-basins are from 1.1 Mm$^3$ to 100.9 Mm$^3$ with a median value of 12.5 Mm$^3$. In each sub-basin, natural river flows and water transferred from the storage facilities (i.e., the agricultural reservoirs and dams) are consumed for agricultural, and municipal and industrial (M&I) purposes. The water diverted for the M&I demands could return to the rivers, becoming available water for lower demand sectors.





For water resources system analyses, we simplified the complex river basin using a node-and-link network as shown in Figure 1. Each sub-basin was conceptualized as a node with natural water availability (i.e., natural runoff), storage capacity (i.e., water storage in the agricultural reservoirs), and water demands (i.e., agricultural and M&I water uses). The sub-basin nodes were connected by the stream links (the continuous lines). The two large dams were represented by the nodes only having storage capacities and located between the two adjacent sub-basins accordingly. The outside water demands sectors were represented by nodes with no natural flows and zero storage capacities. The human-made connections between the dams and the sub-basins are conceptualized by the separate links (the dashed lines) with conveyance limits.

## 2.2 Climatological data

We collected daily precipitation and maximum and minimum temperatures over South Korea at 3-km grid resolution for 1973-2015. The grid data were produced by interpolating synoptic observations at 60 stations in the automated surface observing system (ASOS) operated by the Korea Meteorological Administration. The point weather data were spatially interpolated by the Parameter-elevation Regression on Independent Slope Model (PRISM; Daly et al., 2008), and overestimated values were smoothed by the inverse distance method. Jung and Eum (2015) found improved performance of the combined method in South Korea via comparative evaluations to the original PRISM. For this study, the collected grid data were spatially aggregated with the sub-basin boundaries. For runoff simulations at the sub-basin nodes, we imposed various hypothetical climatic stresses on the spatially aggregated precipitation and temperatures using a semi-parametric stochastic weather generator.

According to the grid data, the mean annual precipitation and temperature over the Geum River Basin for 1976-1995 were 1,245 mm and 11.7 ℃, respectively. They have risen to 1,325 mm (+6.4%) and 12.2 ℃ (+0.5℃) during 1996-2015, providing an indication that atmospheric water supply and demand gradually increase over time. However, climatology within the river basin diverges between the sub-basins. The upper sub-basins at higher elevations tend to get wetter due to increasing precipitation, whereas rising temperatures seem to make the lower basins become drier (Kim, 2018).

## 2.3 Water demand data

The water demands for 2030 were taken as the reference demands to evaluate water supply capacity of the river basin. In South Korea, government-driven national water resources plans are legally developed for sustainable resources management for every bi-decadal period. The water resources plan for 2020 was first established in 2000 including water demand projections up to 2020 (MOCT, 2000), and has been revised three times to consider hydrologic and socioeconomic changes since the initial version (MOCT, 2006; MLTM, 2011; MOLIT, 2016). In the third version of the water resources plan for 2020 (MOLIT, 2016), the water demands across South Korea were re-projected up to 2030. By electronic correspondence (requested on Sep-26/2017), we achieved the demand data projected to 2030 given at 10-day intervals for each sub-basin in the Geum River Basin from the team leading the national water plan at the Korea Institute of Civil Engineering and Building



Technology. We briefly describe about the water demand projections here, and the details are found in the third national plan for 2020 (MOLIT, 2016).

The agricultural and M&I demands are independently projected for the year of 2030 under the high, medium, and low demand scenarios. The agricultural demand within a sub-basin was estimated by the FAO Penman-Monteith equation (Allen et al., 1998). The potential evapotranspiration on each day of year was averaged across the observed periods. The effective rainfall was excluded to represent irrigation requirement. The mean irrigation requirement for each crop was multiplied by the planting area projected for 2030. The planting areas under the medium demand scenario were extrapolated by temporal trends in historical observations. The high demand scenario applied 3.5% additional increase to the demands under the medium scenario, while the low demand scenario decreased them by 3.5%. On the other hand, the M&I demands were projected using historical water use records under hypothetical future growth rates and socioeconomic conditions. The medium scenario assumed 4.0% economic growth and 100% effectiveness of the water conservation programs in the future. For the high and lows demand scenarios, 4.5% and 50% of growth rates and 3.5% and 150% effectiveness of the conservation programs were used, respectively. The three scenarios all expected decreasing water demands from 2016 to 2030 mainly due to declining rice-planting lands. In this study, the high demand scenario was chosen from a conservative perspective. The M&I demands at the four outside nodes were estimated simply by the water transfer records from the two dams for simplicity.

In addition, we collected the information of the minimum flow rates required for ecosystem sustainability, namely instream flows (Jowett, 1997), at seven locations within the river basin. The instream flows are determined by the experts' investigations into water quality and ecological conditions in the vicinity of major rivers in South Korea, and officially announced by the Ministry of Environment and the Ministry of Land, Infrastructure, and Transport (MOLIT, 2016). Though the human water demands (i.e., agricultural and M&I uses) are the first priority of the local and regional authorities in the Geum River Basin (MOLIT, 2016), they are recommended to consider the instream flows in water management. The human demands for the year of 2030 and the environmental requirements are summarized in Table 1. For water resources system modelling in this study, the 10-day demand data were aggregated into monthly values.

## 2.4 GCM projections under greenhouse gas concentration scenarios

We achieved daily projections of 25 GCMs (Table A1) from the archive of the Coupled Model Intercomparison Project Phase 5 (Taylor et al., 2012). Two representative concentration pathways (RCPs), RCP4.5 and RCP8.5, were selected to assess the water supply capacity of the river basin for the upcoming bi-decadal period of 2020-2039. RCP4.5 and RCP8.5 were used as scenarios of stabilized and increasing greenhouse gas concentrations frequently in climate change studies (e.g., Yan et al., 2015; Zhang et al., 2016; Moursi et al., 2017).

The 50 GCM projections (i.e., 25 GCMs × 2 RCPs) were bias-corrected by the de-trended quantile mapping (DQM; Bürger et al., 2013; Eum and Cannon, 2017) that can preserve raw climate change signals given by GCMs. The DQM removes the long-term mean change in projected values first. After applying the ordinary quantile mapping (QM; e.g., Hwang and





Graham, 2013) to the de-trended values, the removed trend is reintroduced to the bias-corrected projections. The de-trending procedure may prevent the exaggeration of raw climate change signals, which is a typical drawback of the ordinary QM. More details about DQM and related bias correction methods are available in Bürger et al. (2013), Cannon et al. (2015), and Eum and Cannon (2017). To correct the 50 GCM projections toward the spatial averaged precipitation and temperatures over

the Geum River Basin, 1976-2005 and 2006-2099 were set as the reference and the projection periods, respectively.

## 3 Methodology

### 3.1 The logistic eco-engineering decision scaling framework

Figure 2 describes the logistic eco-engineering decision scaling framework that combines multiple responses of a hydrologic system to diverse climate exposures. While this concept is similar to the decision scaling (Brown et al., 2012; Steinschneider

et al., 2015a), the eco-engineering decision scaling overlaps the response surfaces of multiple performance metrics to consider conflicting interests of stakeholders in decision-making processes. The size of the climate zone mutually satisfying all performance criteria can measure the system robustness to climatic stressors (Poff et al., 2016). Here, we used the response surfaces of the probability of successive performance instead of the ordinary surfaces. Valid hydrologic models are prerequisites to develop the response surfaces. According to the definition of climate change in IPCC (2007), climatic

perturbations in this study are changes in the bi-decadal mean and variance in observed climatology.

The first step of the bottom-up impact assessment is to generate adequate sets of synthetic weather series perturbed by diverse climatic stresses for an exhaustive sensitivity analysis of system performance. By inputting a set of the generated weather series perturbed by user-defined climate stresses to system models, the system behaviours can be simulated in response to the perturbations. If the stochastic stress tests are repeated with fairly large sets of climatic perturbations, the

relationship between climate stresses and performance metrics of the user's interest can be developed. This relationship, which has been defined as the response surface (Kay et al., 2014), or the climate response function (Brown et al., 2012), were slightly modified by converting the expected performance gained from the stress tests into binary outcomes (i.e., success or failure against the pre-defined thresholds). The binary outcomes can be modelled by the logistic regression. Hereafter, the functions between the binary outcomes and climatic stressors are referred to as the "logistic" response surfaces.

In this study, water supply performance of the study river basin under two possible management polices was assessed using a classical optimization scheme for water resources management. With the optimized decision variables, we evaluated water supply reliabilities at each demands nodes of the conceptualised river basin model. The environmental reliabilities against the instream flows at the seven locations were evaluated together under each management policy. The logistic response surfaces were all overlapped to measure vulnerability of the sub-components to climate stresses from multiple perspectives.

Details are following next.





### 3.2 Generating climate-stress-induced weather series

The stochastic weather generator (WG) by Steinschneider and Brown (2013) was employed to produce plausible daily precipitation and temperature sequences with climatic perturbations. Several bottom-up assessments successfully used this model to evaluate performance of hydrologic systems under varying climate stresses (e.g., Whateley et al., 2014;

Steinschneider et al., 2015b).

Two stochastic models are combined in the semi-parametric WG. The wavelet autoregressive model proposed by Kwon et al. (2007) first generates annual precipitation series spatially-averaged within a region of interest for a desired length (20 years in this study). The wavelet components of the annual precipitation series are extended by the autoregressive model to embed the low-frequency structure inherent in observations. Then, daily weather series conditioned by the random annual

precipitation are simulated by the Markovian bootstrap resampler of Apipattanavis et al. (2007). In this process, the daily observations are resampled by the k-nearest-neighbour scheme and the precipitation occurrence series generated by the standard Markovian process (e.g., Wilks, 1998). The weather data at multiple locations within the region of interest are sampled together for spatial coherence. As the final step, the mean and variance of stochastic precipitation series are adjusted by the ordinary QM to impose climatic perturbations stresses. The temperature series are simply perturbed by adding a

temperature differential. Further in-depth details about the stochastic WG are found in Steinschneider and Brown (2013).

To examine the water supply performance under climatic stresses, we generated 343 sets of precipitation and temperature time series spatially coherent between the 14 sub-basins. The perturbations imposed on the precipitation time series were changes in the mean of non-zero daily precipitation and its coefficient of variance (CV). The mean and CV changes were from -40% to +80% at 20% increments relative to the observations for 1973-2015, respectively (i.e., 7×7 perturbations for

precipitation). The temperature time series were perturbed by adding 0-6 ℃ at 1℃ increments (i.e., 7 perturbations for temperature). The total number of the climatic perturbations is 7×7×7=343. The 343 sets of climate-stress-induced weather series were input to a rainfall-runoff model to quantify natural water flows at the sub-basins.

### 3.3 Simulating natural runoff at the sub-basin nodes

A simple rainfall-runoff model, GR4J (Perrin et al., 2003), was utilized to simulate natural flows at the sub-basins nodes.

GR4J has been frequently adopted for various purposes under diverse climates, such as parameter regionalization (e.g., Oudin et al., 2010), predicting flow durations (e.g., Zhang et al., 2015), and low flow estimations (e.g. Demirel et al., 2015) among many others. The four free parameters of GR4J conceptualize functional behaviours of a watershed in response to lumped precipitation and potential evapotranspiration (PET) inputs. The free parameters implicitly explain soil water storage, groundwater exchange, routing storage, and excess runoff generations within a watershed. The parsimonious structure of

GR4J poses relatively small equi-finality problems in parameter calibration and regionalization (Oudin et al., 2008; Perrin et al., 2007). Perrin et al. (2003) provides the computation procedures in detail.



In this study, a proximity-based regionalization was applied for parameter identification, because almost no natural streamflow observations are available at the outlets of the sub-basins. The operational inflow records at the Yongdam Dam were the only applicable observations for parameter calibration at the sub-basin 3001. For the other sub-basins, the parameter sets were transferred from neighbouring watersheds assessed in Kim et al. (2017). Kim et al. (2017) comparatively

assessed performance of the proximity-based parameter transfer in comparison to several alternative methods, concluding that spatial proximity well captured functional similarity between 45 gauged watersheds in South Korea. The mean Nash-Sutcliffe efficiency (NSE) was 0.53 with a standard deviation of 0.41, when transferring the parameter sets of five neighbouring catchments calibrated with observed hydrographs (Kim et al., 2017). Hence, for the 13 sub-basins from 3002 to 3014, natural flows were simulated with the transferred parameter sets from five nearby gauged watersheds, while flows at

the sub-basin 3001 were generated by the parameters calibrated against the inflow data. The five runoff simulations were averaged for the sub-basins in which the regionalization scheme was used. The parameter set calibrated against the inflow records during 2007-2015 for the sub-basin 3001 yielded a NSE value of 0.62. The daily natural flows simulated by GR4J with the 343 stochastic weather sets were temporally aggregated at monthly values for water resources system analyses.

### 3.4 Evaluating water supply and environmental reliabilities

The total water availability in the river basin during a certain month is water storages in the dams and reservoirs at the end of the previous month plus the natural flows at the sub-basins in the current month. Some of the available water is again kept in the storage facilities for supplying water in upcoming months. Thus, operators' decisions on water storages in each month recursively affect supply performance in the river basin through a bi-decadal period. A monthly sequential optimization model was used to determine amounts of the water storages and uses at each sub-basin. The operators should minimize water

deficiency during a current month, while the water storages need to be maximized for water availability in upcoming months. We assumed that the two conflicting objectives are equally important for the operators. Hence, the objective function to determine water supplies and storages at the nodes for a particular month was:

$$\text{Minimize} \quad \frac{\sum D_i - \sum S_i}{\sum D_i} - \frac{\sum V_i - \sum C_i}{\sum C_i} \qquad (1a)$$

$$D_i = DA_i + DM_i \quad (1b)$$
$$S_i = SA_i + SM_i \quad (1c)$$

where, $D_i$ is the total demand, $S_i$ is the total supply, $V_i$ is the water storage, and $C_i$ is the storage capacity ($C_i$) at node i. $SA_i$ and $SM_i$ are agricultural and M&I water supplied for agricultural ($DA_i$) and M&I demands ($DM_i$) at node i, respectively. The

total water demand at each node ($D_i$) is the sum of agricultural demand ($DA_i$) and M&I demand ($DM_i$). Likewise, the water supply at each node is divided into agricultural supply ($SA_i$) and M&I supply ($SM_i$).



The monthly optimizations were subject to constraints. The water supply ($S_i$) to a demand node was limited by water availability, which is the sum of natural flow at the node in the current month, flows from other nodes via the stream and the human-made links in the current month, and water storage at the node in the previous month. Water surplus at the nodes was not allowed (i.e., $S_i \leq D_i$). The water remained after supplies and storage at a sub-basin node should be discharged from the node through the channel network. The water storage at each node is constrained by its storage capacity ($V_i \leq C_i$). The water transfers through the human-made links were only supplied for M&I demands of destination nodes, and were limited by the conveyance capacity (40 $Mm^3$ $month^{-1}$). The agricultural and M&I demands were of equal priorities in optimizations.

Using 20-year-long natural flows per climatic perturbation, we determined $SA_i$, $SM_i$, and $V_i$ month by month using the global optimization tool "fmincon" in the Matlab software. Since $V_i$ values determined for a month become water availability for its next month, optimizations for 240 months interplay sequentially. To consider the return flows, we followed the hypotheses in the water plan for 2020 (MOLIT, 2016). Simply, 65% of the M&I water use at each node was assumed to return and become available water for following nodes, while no return flows after agricultural uses were considered in the water plan due to high water use efficiency.

Using the optimized $SA_i$, $SM_i$, and $V_i$ values, water supply performances at the demand nodes were measured for the given 20-year-long stress-imposed weather series. For each demand node of the node-and-link network, we measured the water supply reliability ($\rho_{s,i}$) defined as the probability of satisfactory supply against 99% of the demand:

$$\rho_{s,i} = prob\ [S_i \geq 0.99 D_i] \qquad (2)$$

The amount of water passing the seven locations with the instream flow requirements can be also calculated using the decision variables and the natural flows. The environmental reliability at location j ($\rho_{e,j}$) was evaluated by:

$$\rho_{e,j} = prob\ [F_j \geq F_{min,j}] \qquad (3)$$

where, $F_j$ and $F_{min,j}$ are the flow passing location j and the instream flows required for ecosystem sustainability at the location, respectively.

In total, reliabilities at 18 demand nodes (14 sub-basins and 4 demand nodes out of the basin) and 7 locations requiring the environmental flows were evaluated for each climatic perturbation. These 25 performance indicators and corresponding climatic perturbation were used to develop response functions of system performance to climatic stressors.

**3.5 Developing the logistic response surfaces**

The logistic regression analyses were used to identify responsive behaviours of system performance to bi-decadal climatic stresses. With no needs of the homogeneity and normality assumptions, this regression model can explain occurrences of a





probabilistic process with relevant independent variables by assuming a linear relationship between the logit of the explanatory variables and binary responses (e.g., Lee et al. 2016). We categorized the water supply and environmental reliabilities gained from the stress tests into binary outcomes (i.e., satisfactory or non-satisfactory against a threshold). The binary outcomes corresponding to the 343 climatic perturbations were explained by the logistic equation:

$$\pi = \frac{1}{1+\exp[-(\beta_0+\beta_1 X_1+\beta_2 X_2+\cdots)]} \qquad (4)$$

where, $\pi$ is the probability of successive outcomes, $\beta_i$ are the regression coefficients, and $X_i$ are explanatory variables.

Each weather series generated by the WG could be summarized with three bi-decadal climatic properties of the mean annual

precipitation ($P_{avg}$), the CV of daily precipitation ($P_{cv}$), and the mean annual temperature ($T_{avg}$). $P_{avg}$, $P_{cv}$, and $T_{avg}$ were used as the candidate explanatory variables.

In our sensitivity analysis, the climatic bounds were found at $\pi=95\%$ for the sub-components of the study river basin. We combined all the boundaries developed by the logistic regressions into a single climate domain. The GCM projections for 2020-2039 were overlaid on the domain to assess impacts of greenhouse gas emissions on the water supply and

environmental reliabilities.

## 4 Results

### 4.1 Evaluating water supply reliability using the logistic response surfaces

Figure 3 displays the scatter plots between the three explanatory variables ($P_{avg}$, $P_{cv}$, and $T_{avg}$) and water supply reliability ($\rho_s$) at the sub-basin 3001 collected from the 343 sets of sequential optimizations. We preliminarily checked statistical

significance of the explanatory variables using the linear multiple regression. $P_{avg}$ was the most significant to explain variation of the $\rho_s$ values (p-value $< 10^{-16}$) followed by $T_{avg}$ (p-value $< 10^{-13}$), whereas $P_{cv}$ was of a low significance below 5% level (p-value = 0.28). This implies that the bi-decadal water supply reliability generally determined by variations of the mean precipitation and the mean temperature. Though higher precipitation variability ($P_{cv}$) indicates more intensified rainfall events generating larger direct runoff, storage capacities of the agricultural reservoirs and the dams seem to dampen the

effects of $P_{cv}$ changes on the variation of $\rho_s$. The preliminary regression analysis led us to an indication that the two explanatory variables $P_{avg}$ and $T_{avg}$ could sufficiently capture the variation of water supply performance across the demand nodes.

The linear regression between ($P_{avg}$, $T_{avg}$) and the $\rho_s$ values can show the responses of a system performance to climatic stresses. Figure 4a illustrates the regression surface between $\rho_s$ and $P_{avg}$ and $T_{avg}$ changes relative to 1996-2015 ($R^2 = 0.90$) at

the sub-basin 3001, on which the collection of 50 GCM projections was overlaid. All of the 50 GCMs expected that $\rho_s$ at the sub-basin would be greater than 0.95 for 2020-2039. This type of response surfaces between the expected performance and





hypothetical climatic stresses have been commonly used in the bottom-up assessments (e.g., Brown et al., 2012; Whateley et al., 2014; Turner et al., 2014). On the other hand, the logit function fitted to the binary outcomes categorized by a threshold enables to estimate the probability that system performance would be successive. As shown in Figure 4b, we developed the logistic response surface using the 343 binary outcomes against a threshold of $\rho_s = 0.95$ and corresponding $T_{avg}$ and $P_{avg}$

values for the sub-basin 3001 (McFadden $R^2 = 0.84$). The 50 individual GCMs predicted the probability of $\rho_s > 0.95$ (hereafter referred to as $\pi_{s95}$) with a range of 78-99% for 2020-2039. A lower $\pi_{s95}$ implies a higher risk that the optimal water management would yield unsatisfactory supply reliability of $\rho_s \leq 0.95$. Though the GCMs overlaid on the ordinary response surface (Figure 4a) seemingly expect 0.95 or greater $\rho_s$ for 2020-2039, the logistic response surface (Figure 4b) implies that $\pi_{s95}$ would decrease from 99% to 95% with the same climate projections. It should be noted that the climatic bound for $\rho_s >$

0.95 in the ordinary response surface (dotted lines in Figure 4) corresponds to only 45-65% of $\pi_{s95}$. Hence, a user of the logistic response surface may perceive a higher climate change risk than those who gauge system performance with the ordinary response surface.

For each demand node, we developed the logistic response functions of the binary outcomes against $\rho_s = 0.95$ using the sequential optimization results, and found the climatic bounds at which $\pi_{s95}$ was 95%. Figure 5 shows the climatic bounds of

$\pi_{s95} = 95\%$ for the all demand nodes together. The climatic bounds for the demand nodes were all lower than the bound of $\pi_{s95} = 95\%$ for the total water demands in the entire basin, indicating that operations toward minimizing overall water deficiency across the river basin may not force local unsatisfactory performance at the demand nodes. However, vulnerability to climatic stresses differs between the demand nodes. The sub-basin 3001 (the uppermost demand node) seems to be exposed to the highest climate change risks. It had the smallest climate zone for 95% or higher $\pi_{s95}$, and the $P_{avg}$ range

for $\pi_{s95} > 95\%$ sensitively declined with rising $T_{avg}$. The second vulnerability was found at the external demand node receiving water from the Yongdam Dam, which stores discharges from the sub-basin 3001. On the contrary, the lowermost node (sub-basin 3014) has the largest climate zone for $\pi_{s95} > 95\%$. Even with 40% reduction in $P_{avg}$, the sub-basin 3014 would be able to meet the threshold of $\rho_s = 0.95$ with 95% or a higher $\pi_{s95}$. Its $P_{avg}$ range for $\pi_{s95} > 95\%$ was almost insensitive to rising $T_{avg}$. This is because the lower sub-basins below the Daecheong Dam could receive streamflow

discharged from the upper sub-basins; thereby, they could withstand much stronger climatic stresses despite the high agricultural demands.

## 4.2 Evaluating reliability against the instream flow requirements

The monthly flows at the seven locations requiring the instream flows were calculated with the decision variables obtained from the optimizations and the given natural flow estimates. We compared the flows against the minimum requirements, and

evaluated the environmental reliabilities (i.e., $\rho_e$) at each location. Figure 6 shows the box plots of $\rho_e$ values at the seven locations in response to the 343 climate perturbations. While $\rho_e$ values at all the locations decreased as climate became drier, the location E seems to be the most vulnerable. Even with no changes in $P_{avg}$ and $T_{avg}$, the outflows from the sub-basin 3011 were often less than the minimum requirement, implying that ecosystems near the location E might be currently undermined





by the large agricultural and M&I demands at the sub-basin 3011. If $P_{avg}$ was declined by 20% and $T_{avg}$ rose by 3°C, $\rho_e$ at the location E would fall below 0.5. On the other hand, streamflow at the location D perfectly satisfied the minimum requirement under the same stress. Despite the second-largest M&I demands at the sub-basin 3009, water transfers from the two large dams could deliver sufficient water supplies. 65% of the water supplies for M&I demand was supposed to return to

the stream network and became streamflow to meet the instream flow requirement at the location D. Although both sub-basins 3009 and 3011 have limited geomorphological connectivity to the upper sub-basins, their demand components and linkages to the two large dams made the significant difference between their $\rho_e$ values.

We developed the logistic response surfaces with binary outcomes against $\rho_e = 0.70$ and corresponding ($T_{avg}$, $P_{avg}$) pairs. The threshold $\rho_e = 0.70$ was selected for satisfactory performance at the location E under no climate perturbations (i.e., no

changes in $P_{avg}$ and $T_{avg}$ relative to 1996-2015). Figure 7 displays the climatic bounds at which the probability of $\rho_e > 0.70$ is 95% (hereafter referred to as $\pi_{e70}$) As expected from Figure 6, the bound for the location E was the highest, and the climate zone for $\pi_{e70} > 95\%$ sensitively declines with rising $T_{avg}$. The bound for $\pi_{s95} = 95\%$ at the sub-basin 3001 (orange dashed line) was below the bounds for the locations E and F. The human-demand-only operations would increase the risks of ecosystem degradation near the locations E and F if climate alters toward drier conditions. The environmental risks at the locations E

and F seem to be more sensitive to rising $T_{avg}$ than the water supply risk at the sub-basin 3001. The 50 GCM projections for 2020-2039 predicted that $\pi_{e70}$ at the location E would drop by 32% relative to 1995-2015. Only 6 out of the 50 GCMs (12%) fell within the range mutually satisfying $\pi_{s95} > 95\%$ and $\pi_{e70} > 95\%$ for every demand node and for the all locations requiring the minimum flows.

### 4.3 Considering the instream flows into water management

From the assessment with the logistic response surfaces and the GCM projections, the environmental risk for 2020-2039 was likely to become an issue due to climate change. As an adaptation strategy, the instream flows could be considered in water management to be balanced between water supply and environmental risks. We modelled this scenario by including the environmental requirement in the objective function for the sequential optimizations as:

Minimize $\frac{\sum D_i - \sum S_i}{\sum D_i} - \frac{\sum V_i - \sum C_i}{\sum C_i} + w \frac{Q_{min,E} - Q_E}{Q_{min,E}}$      (4)

where, $Q_{min,E}$ and $Q_E$ are the instream flow requirement and the streamflow at the location E respectively. w is a weight representing the relative importance of flow deficiency against the environmental requirement. While the minimum instream flow could be treated as a constraint for optimizations, this approach may lead to no optimal solutions under severe climate

stresses. Hence, it would be better to consider the environmental requirement in the objective function for varying climate stresses. Through trial and error experiments with the 343 climate perturbations, we found that $\rho_e$ at the location E could



substantially increase with a small w value. The sequential optimizations with w=0.01 allowed us to have a fairly improved $\rho_e$ value at the location E.

The climate bounds in Figure 8 summarize the two scenarios with and without the environmental requirement. By comparing the middle panels, it is indicated that considering the minimum flow in the objective function significantly lowered the climatic bound of $\pi_{e70} = 95\%$ at the location E, thereby widening the climatic zone within which all the locations mutually satisfy $\pi_{e70} > 95\%$. However, in trade-off, the bound of $\pi_{s95} = 95\%$ for the sub-basin 3002 moved upward, narrowing the climatic zone mutually satisfying $\pi_{s95} > 95\%$ for all the demand nodes. Overall, the climatic zone mutually satisfying $\pi_{s95} > 95\%$ and $\pi_{e70} > 95\%$ for all the sub-components was expected to decrease when the minimum instream flow at the location E was considered. Based on reduction in the mutual climate zone, considering the environmental requirement may be unattractive to operators in the Geum River Basin due to the higher priority in human demands.

The decreasing $\pi_{s95}$ at the sub-basin 3002 could be explained by considering that the instream flow required at the location E should force water retained in the sub-basin 3011 to be released even for low-demand seasons. Since the early water release from the sub-basin 3011 should reduce water availability during rice-planting seasons for the sub-basins 3012 and 3014 requiring large agricultural demands, more water resources need be transferred from the dams and the upper sub-basins. Consequently, it was inevitable to have deficient water supplies in the sub-basin 3002 with relatively small demands even under optimized water allocations.

The impact assessments using the logistic bounds show that the sub-basins 3001 and 3002 would be the most vulnerable to climate stresses for water supply. The highest environmental risks from climate change would be found at the locations E and F against the instream flow requirements. The box plots in Figure 9 display $\pi_{s95}$ and $\pi_{e70}$ expected by the 50 GCM collections at the two sub-basins and the two environmental locations. The 50 GCMs projections for 2020-2039 expected almost 100% of $\pi_{s95}$ at the sub-basins 3002 when the instream flow was not considered. For this extremely low risk in water supply reliability, operators should take a substantial risk of ecosystem degradation in the vicinity of the location E. To reduce the environmental risk, on the other hand, fairly increasing risks of $\rho_s \leq 0.95$ at the sub-basin 3002 should be accepted. The water supply and environmental reliabilities at the sub-basin 3001 and the location F seems to be minimally affected by the minimum flow requirement at the location E.

## 5 Discussion and conclusions

### 5.1 The logistic response surfaces for impact assessment of climate change

The ordinary response surface allowed users to estimate the expected performance of hydrologic systems in response to climate stressors, providing convenience to promptly evaluate many GCM projections (e.g., Steinschneider et al., 2015a; Brown et al., 2012; Prudhomme et al., 2010). Nevertheless, the response surface of performance metrics might provide insufficient probabilistic information needed for decision-making processes. It is possible to count the climate projections satisfying a desired threshold from a collection of GCMs using the ordinary response surface (e.g., Moursi et al., 2017).




However, the GCM counts do not properly represent the probability of successive (or non-successive) performances because it highly depends on the size of the GCM collections. Even in the case that all GCMs in users' hand fall within the desirable climate zone above the threshold, it would not imply that future system performance would be successive at 100% confidence.

The logistic response surface could supplement this weakness of the ordinary climate response function. It enables to directly estimate the probability of successive performance from a single climate projection, allowing users to perceive potential risks of system failures associated with the given projection. The probability estimates from the logistic response function can play a role in risk-based decision-making particularly when an adaptive solution should be targeted at reference climate conditions (e.g., resizing infrastructures for a reference climate condition). If the ordinary and the logistic responses surfaces

together, decision-makers can more clearly understand the probability of successive (or non-successive) outcomes together with the expected performance under varying climatic stresses.

## 5.2 Eco-engineering decision scaling with the logistic climate bounds

The eco-engineering decision scaling (Poff et al., 2016) integrates the responsive behaviours of multiple performance metrics into a single climate space. It may lead decision-makers to adaptive solutions that can widen the climate zone

mutually satisfying conflicting criteria. In Poff et al. (2016), a prominent example was provided to assess adaptation costs and potential environmental risks in order to find the most robust adaptation strategies for a dam site. In this study, we showed the advantage of this framework to explore performances at sub-components across a large system in a single climate domain.

One difference between our approach and Poff et al. (2016) is that we used the logistic climate bounds to measure the size of

the climate zone mutually satisfying the risk thresholds. The original eco-engineering decision scaling was focused on the expected system performance, since the ordinary response surfaces were used. Then, a question can be raised as to "what if no climate projections fall within the mutual climate zone?" In this case, stakeholders may perceive unacceptable risks even though the adaptive solutions can enlarge the mutual climate zone for multiple criteria. On the other hand, if the logistic response surfaces were employed instead, users can evaluate the risk of non-successive outcomes from each climate

projection as well as the mutual zone representing the system robustness. The logistic eco-engineering decision scaling can simultaneously show how much system robustness can be obtained from an adaptive solution and how much risks of system failures are indicated by climate projections. Hence, the logistic decision scaling may be more informative for decision-making processes.

## 5.3 Limitations

There are several caveats in this study. First, the monthly operations seeking a balance between water scarcity and storages might be only a part of operators' interests. The stakeholders in practice may be subject to other practical constraints (e.g.,



reducing flood risks or hydropower generations) rather than taking actions toward optimal water supply efficiency. Thus, this study should be deemed a special case focused on the maximum supply reliability under the given operational objectives.

Second, the climate bounds of π=95% used in this study indicate the climatic stresses under which the sub-systems of the river basin can successively perform at 95% confidence. These limits are unnecessarily decision thresholds, but indicators of

potential risks. To determine the decision thresholds, the ordinary response surface may be more useful. The logistic response surfaces may be suitable after stakeholder-driven thresholds are defined.

Finally, given the substantial uncertainty sources associated in the stochastic stress tests (Steinschnieder et al., 2015b), the logistic regressions may be an oversimplified approach to assessing the risk of system failures when a small number of stochastic realizations is used in the stress tests. More reliable risk estimates can be achieved from other uncertainty

assessment methods though expensive efforts may be required.

## 5.4 Conclusions

In this study, we proposed to combine the logistic regressions with the eco-engineering decision scaling framework to evaluate the risk of system failures instead of the expected performance. The case study shows that the bottom-up impact assessments could be more probabilistically translated by simply converting the expected performance from the stress tests

into the binary outcomes against pre-defined thresholds. In the case that a narrow satisfactory zone was found in the ordinary response surfaces, decision-makers may need to predict the risk of system failures rather than the expected performance. The logistic response surfaces may be a practical approach that requires minimal computational costs for the risk assessment with given climate projections. From the application of the proposed framework, following conclusions are worth emphasizing:

(1) The logistic response surface can provide convenience to explore potential risks of unsatisfactory outcomes against a pre-

defined performance threshold in response to climatic stresses, while the ordinary response surface can show the expected system performance only.

(2) The eco-engineering decision scaling can be flexibly used to evaluate performance at sub-components of a complex hydrologic system. By integrating responsive behaviours of all the sub-components into one single climatic space, users can easily find the most vulnerable sub-component to climate stressors.

(3) The logistic eco-engineering decision scaling may be more informative than the original framework for risk-based decision-making processes. The potential risks of unsuccessful outcomes contained in adaptive solutions can be evaluated from a single climate projection in addition to the mutual climate zones indicating system robustness to climatic changes.

(4) The case study for the Geum River Basin in South Korea provides an assessment that water supply performance for 2020-2039 seems to be sufficient against the water demands projected to 2030. However, the human-demand-only

operations would make the eco-systems increasingly vulnerable. To consider the instream flow requirement in operations for 2020-2039, risks of insufficient water supply should increase at the upper sub-basins with small water demands.





**Acknowledgement**

This study was supported by the APEC Climate Center. We are very thankful for the water demand data provided by the team leading for the national water resources plan at the Korea Institute of Civil Engineering and Building Technology. The GCMs downscaled by Dr. Hyung-Il Eum at the Alberta Environment and Parks are greatly appreciated. All authors declare
no conflict of interests. The data required to reproduce the results are available upon request from the authors (d.kim@apcc21.org, sjchoi@kict.re.kr).

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





## Appendix 1. Information of the General Circulation Models collected from the CMIP5 archive.

Table A1. List of the selected GCMs for the impact assessment under the RCP 4.5 and 8.5 scenarios.

| No. | Model Name | Resolution (degree) | Producing Institution |
|---|---|---|---|
| 1 | CMCC-CM | 0.750×0.748 | Centro Euro-Mediterraneo per I Cambiamenti Climatici |
| 2 | CCSM4 | 1.250×0.942 | National Center for Atmospheric Research |
| 3 | CESM1-BGC | 1.250×0.942 | |
| 4 | CESM1-CAM5 | 1.250×0.942 | |
| 5 | MRI-CGCM3 | 1.125×1.122 | Meteorological Research Institute |
| 6 | CNRM-CM5 | 1.406×1.401 | Centre National de Recherches Meteorologiques |
| 7 | HadGEM2-AO | 1.875×1.250 | Met Office Hadley Centre |
| 8 | HadGEM2-CC | 1.875×1.250 | |
| 9 | HadGEM2-ES | 1.875×1.250 | |
| 10 | INM-CM4 | 2.000×1.500 | Institute for Numerical Mathematics |
| 11 | IPSL-CM5A-MR | 2.500×1.268 | Institut Pierre-Simon Laplace |
| 12 | MPI-ESM-LR | 1.875×1.865 | Max Planck Institute for Meteorology (MPI-M) |
| 13 | MPI-ESM-MR | 1.875×1.865 | |
| 14 | FGOALS-s2 | 2.813×1.659 | LASG, Institute of Atmospheric Physics, Chinese Academy of Sciences |
| 15 | NorESM1-M | 2.500×1.895 | Norwegian Climate Centre |
| 16 | GFDL-ESM2G | 2.500×2.023 | Geophysical Fluid Dynamics Laboratory |
| 17 | GFDL-ESM2M | 2.500×2.023 | |
| 18 | BCC-CSM1-1 | 2.813×2.791 | Beijing Climate Center, China Meteorological Administration |
| 19 | BCC-CSM1-1-M | 1.125×1.122 | |
| 20 | IPSL-CM5A-LR | 3.750×1.895 | Institut Pierre-Simon Laplace |
| 21 | IPSL-CM5B-LR | 3.750×1.895 | |
| 22 | MIROC5 | 1.406×1.401 | Atmosphere and Ocean Research Institute, National Institute for Environmental Studies, and Japan Agency for Marine-Earth Science and Technology |
| 23 | MIROC-ESM-CHEM | 2.813×2.791 | |
| 24 | MIROC-ESM | 2.813×2.791 | |
| 25 | CanESM2 | 2.813×2.791 | Canadian Centre for Climate Modelling and Analysis |



**Table 1: Annual agricultural and M&I demands per demand node and the minimum instream flows from the sub-basins corresponding to the seven locations.**

| | ID No. | Mean annual flow[*] ($Mm^3\ yr^{-1}$) | Agricultural demand ($Mm^3\ yr^{-1}$) | M&I demand ($Mm^3\ yr^{-1}$) | Total storage capacity ($Mm^3$) | Instream flow requirement ($Mm^3\ month^{-1}$) |
|---|---|---|---|---|---|---|
| Sub-basin node | 3001 | 639.6 | 50.9 | 7.6 | 29.7 | |
| | 3002 | 97.3 | 4.0 | 0.4 | 1.0 | |
| | 3003 | 254.5 | 13.1 | 2.4 | 5.0 | |
| | 3004 | 498.6 | 50.5 | 15.8 | 10.1 | 8.9 (A) |
| | 3005 | 382.8 | 42.3 | 5.1 | 14.9 | 6.6 (B) |
| | 3006 | 82.6 | 9.2 | 3.3 | 6.8 | |
| | 3007 | 384.0 | 74.2 | 6.1 | 22.0 | 7.4 (C) |
| | 3008 | 473.6 | 36.9 | 34.4 | 7.2 | |
| | 3009 | 465.8 | 31.5 | 208.5 | 7.0 | 6.7 (D) |
| | 3010 | 92.2 | 19.3 | 16.4 | 0.2 | 20.8 (E) |
| | 3011 | 1145.0 | 356.6 | 296.2 | 100.9 | |
| | 3012 | 1437.2 | 367.9 | 75.1 | 38.4 | 45.9 (F) |
| | 3013 | 506.1 | 193.2 | 26.4 | 47.4 | 6.4 (G) |
| | 3014 | 340.7 | 215.1 | 10.3 | 29.6 | |
| Outside demand node | OD 1 | | | 20.6 | | |
| | OD 2 | | | 42.1 | | |
| | OD 3 | | | 5.1 | | |
| | OD 4 | | | 4.0 | | |
| Total | | 6,800.0 | 1464.7 | 779.8 | 320.2 | |

[*] Natural runoff averaged over 20-year rainfall-runoff simulations with stochastic weather series containing zero climatic perturbations ($\Delta P_{avg} = 0\%$, $\Delta P_{cv} = 0\%$, $\Delta T_{avg} = 0\%$) relative to 1996-2015.



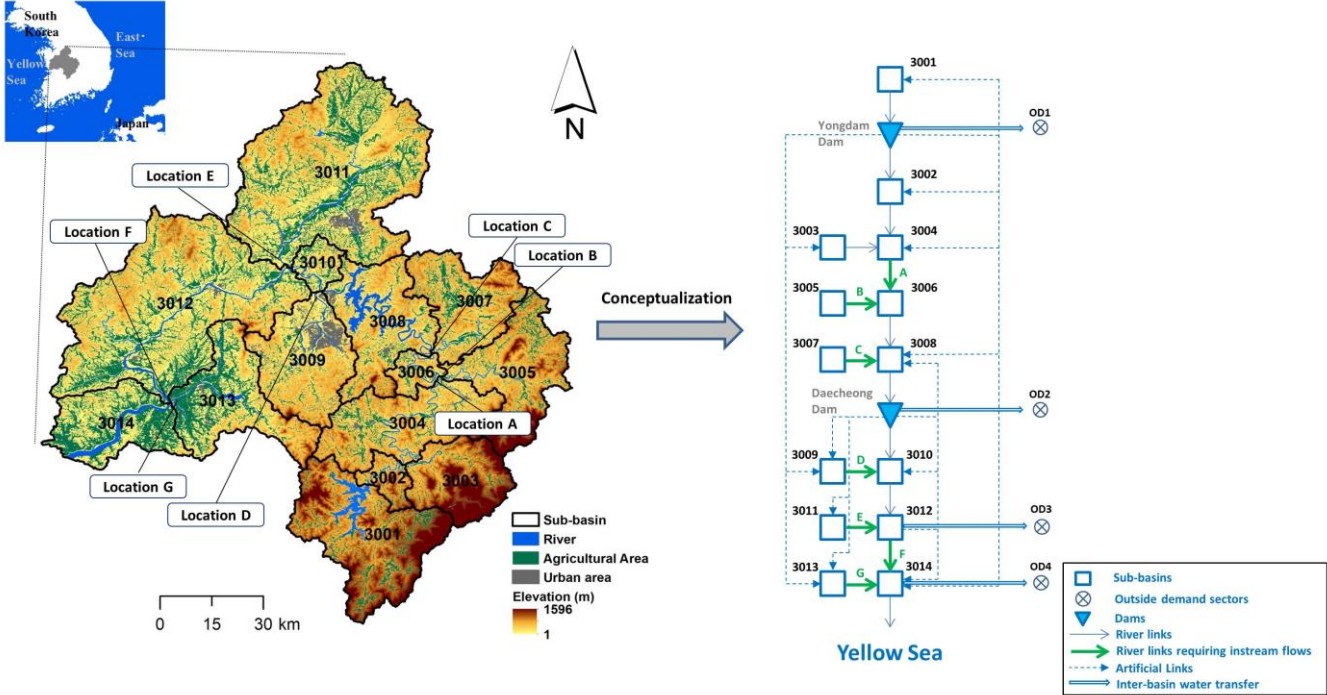

Figure 1: Layout of the Geum River Basins (left) and the simplified node-and-link network for modelling water allocations (right).



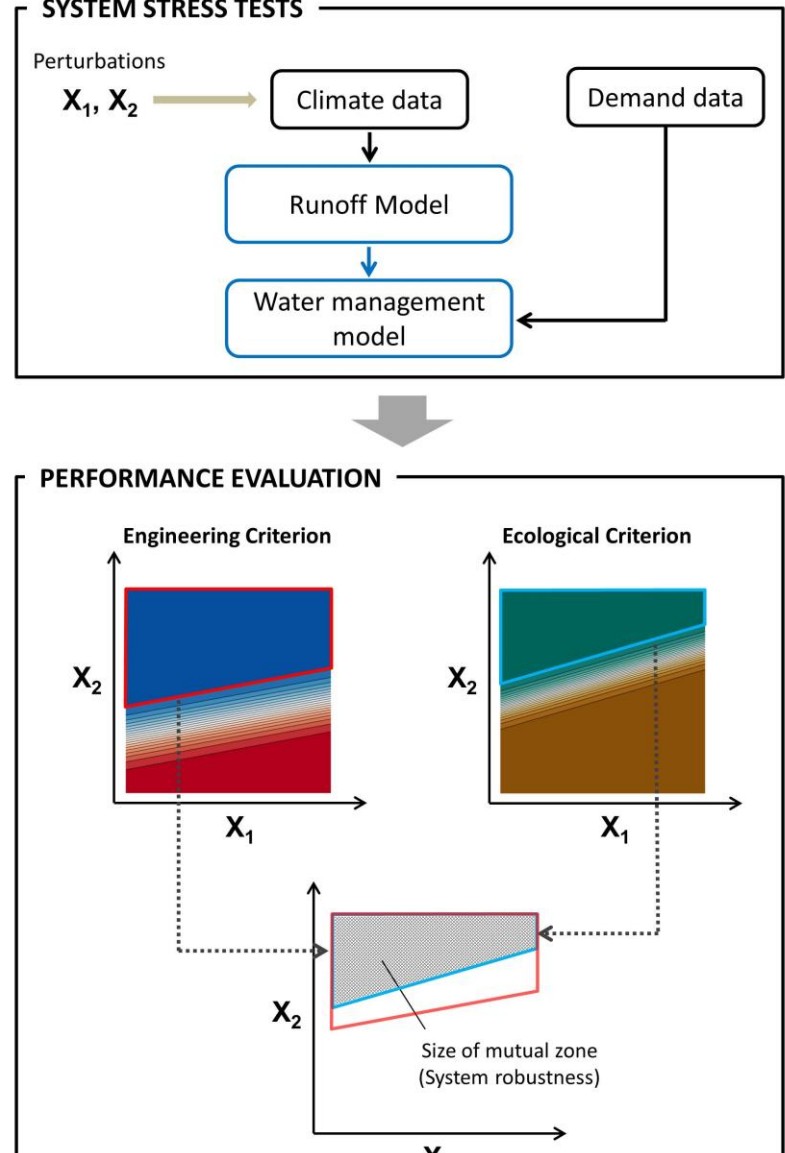

**Figure 2: The schematic of the logistic eco-decision scaling framework for evaluating climate change risks with multiple performance criteria.**




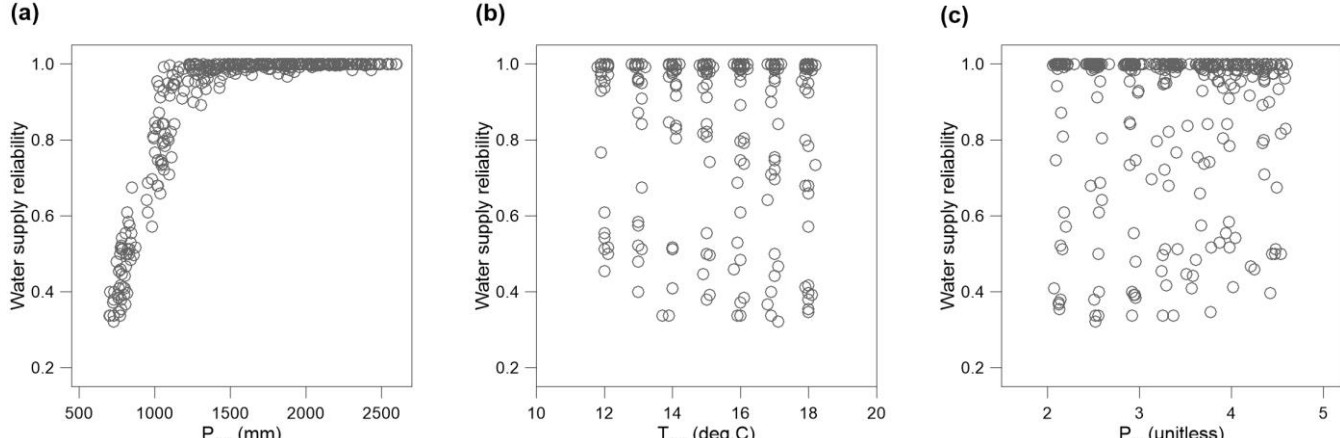

**Figure 3: Scatter plots between the water supply reliability samples for the sub-basin 3001 and corresponding (a) $P_{avg}$, (b) $T_{avg}$, (c) $P_{cv}$ collected from the stress tests with the 343 sets of the stochastically generated stress-imposed weather series.**





**Figure 4: (a) Response surface of $\rho_s$ at the sub-basin 3011 to changes in $P_{avg}$ and $T_{avg}$, (b) response surface of probability that $\rho_s$ is greater than 0.95. The red arrows indicate climatic change to the ensemble of the 50 GCM projections for 2020-2039. The dashed lines are the climatic bounds above which the expected $\rho_s$ would be greater than 0.95. The bottom panels are the scatter plots between $P_{avg}$ changes and supply reliability (left) and binary outcomes (right) used for the response surfaces, respectively. The changes in $P_{avg}$ and $T_{avg}$ are relative to observed climatology for 1996-2015.**





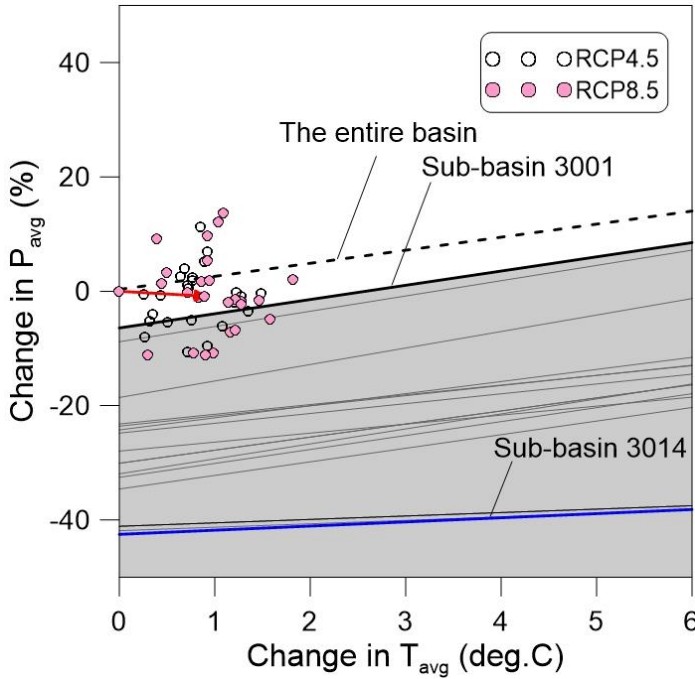

**Figure 5: Climatic bounds for $\pi_{s90} = 95\%$ for each demand node and the entire basin, on which the 50 GCMs were superimposed. The symbols are the 50 GCMs projections, and the red arrow indicates climatic change to the ensemble of the GCMs for 2020-2039.**

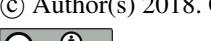



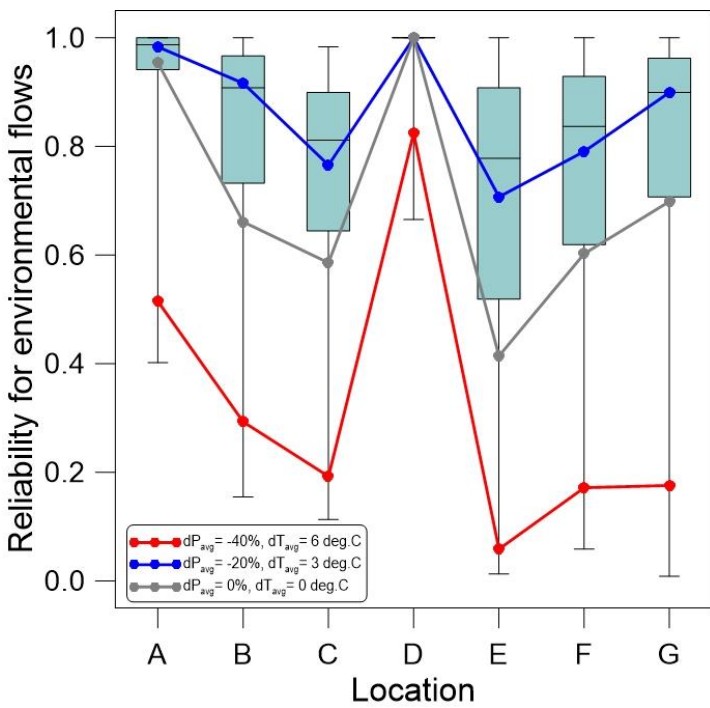

**Figure 6: Reliability against the instream flow requirements at the seven locations obtained from the 343 sequential optimizations with stress-induced weather series. The blue, grey, and red lines connect the reliabilities at the seven locations under the three representative climatic stresses.**




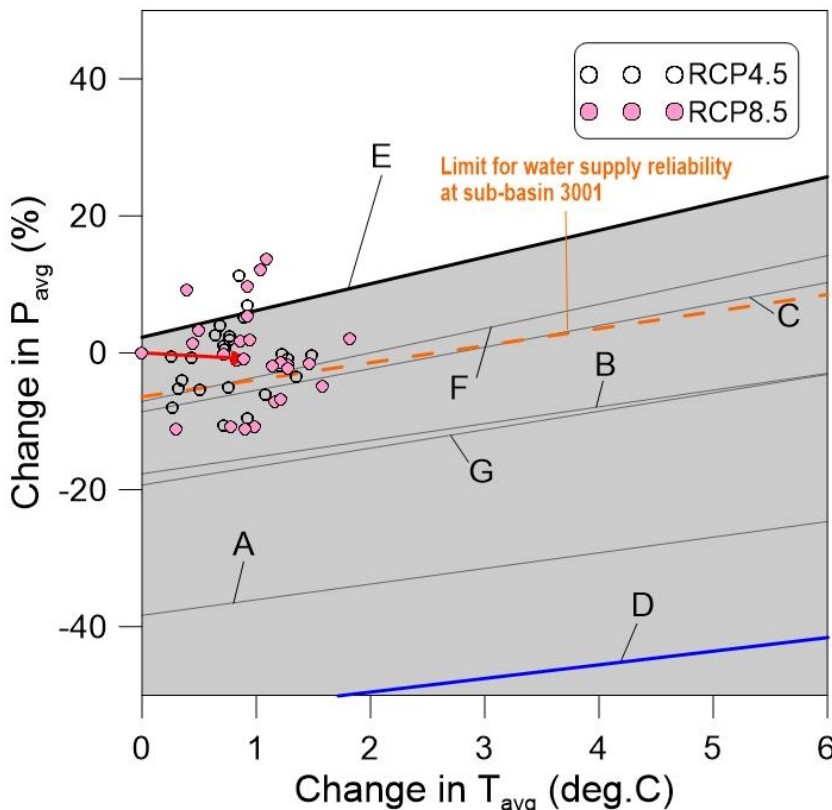

**Figure 7: As in Figure 5, but for $\pi_{e70} = 95\%$ at the seven locations requiring the instream flows for ecosystem sustainability.**



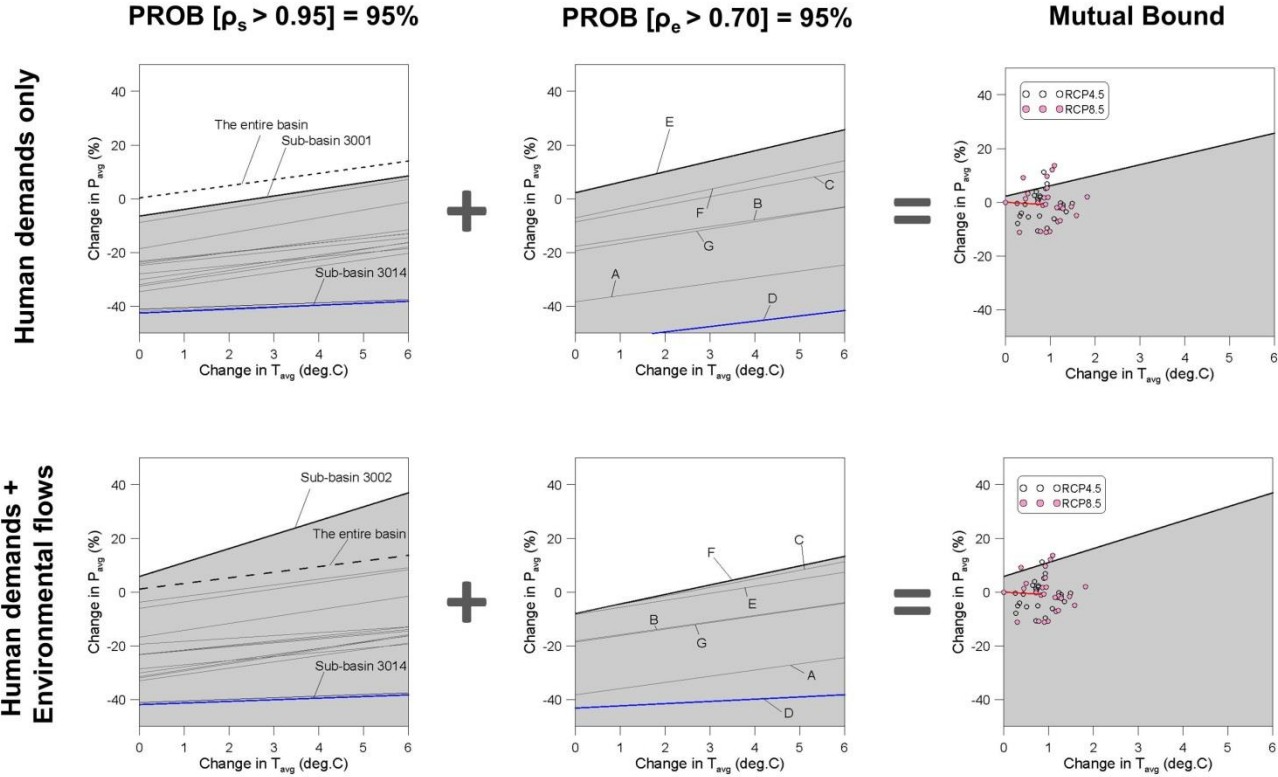

**Figure 8: Climatic bounds for $\pi_{s95}$ = 95% (left) and $\pi_{e70}$ = 95% (middle), and the bound mutually satisfying both criteria (right) under the human-demands-only operations (top) in comparison to the operations considering the instream flows at the location E (bottom). The symbols on the right panels are the 50 GCM projections and the arrows indicate climate change to the ensemble of the GCMs for 2020-2039.**





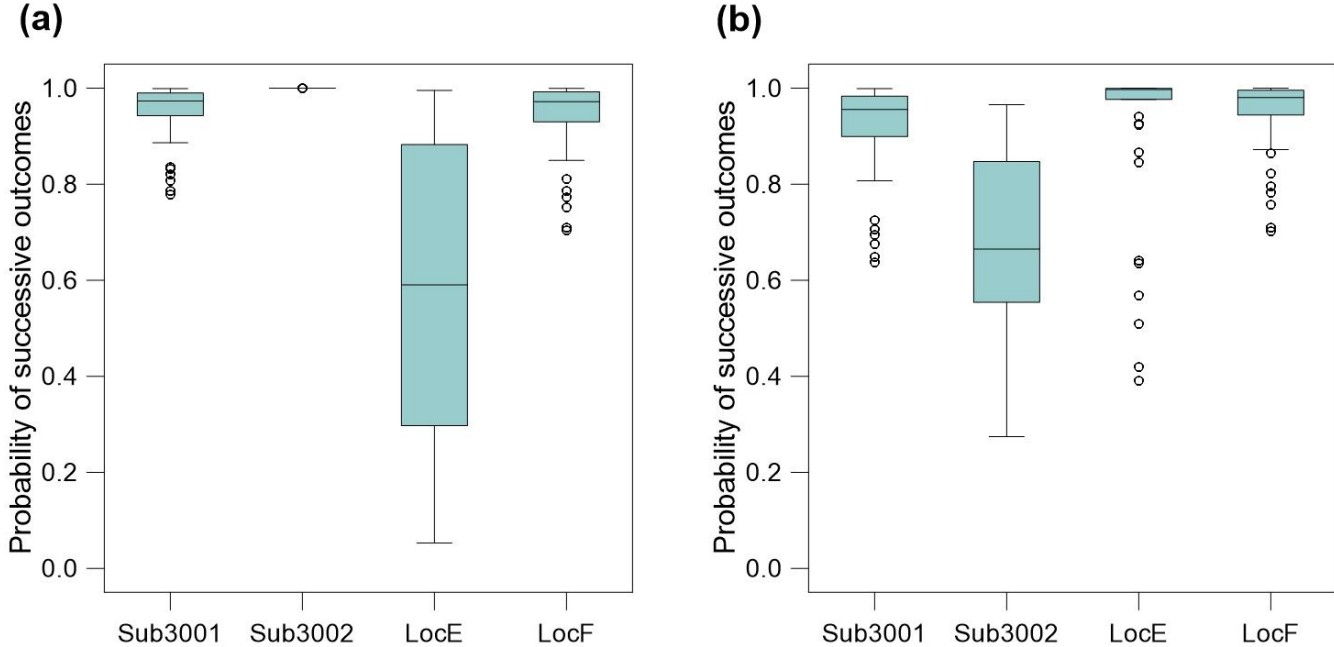

**Figure 9: Box plots of $\pi_{s95}$ and $\pi_{e70}$ estimated from the 50 GCM projections for 2020-2039 (a) with the human-demands-only operations and (b) with the operations considering the instream flow requirement at the location E.**