# Peer review of "Assessing water supply capacity in a complex river basin under climate change using the logistic eco-engineering decision scaling framework"

_Hydrology and Earth System Sciences, 2018_

## Referee Comment (RC1) · Anonymous Referee #1 · 17 Jul 2018

"Assessing water supply capacity in a complex river basin under Climate change using the logistic eco-engineering decision scaling Framework" Authors: Daeha Kim, Jong Ahn Chun, Si-Jung Choi Journal: Hydrology and Earth System Sciences (HESS) Recommendation: Minor Comment

General Comments: This paper has integrated the logistic regressions with the eco-engineering decision scaling framework to evaluate the risk of system failures in contrast to expected performance under dynamic climate change scenarios. This paper contains new insights and contains a lot of information for scientific community. However, the authors have explained the manuscript in complicated ways which I think

could be explained in a simplified manner therefore, the authors are advised to avoid using complex English sentences and try to make their next manuscript as simple as possible which would ultimately increase understanding as well as attract more readers. In a nutshell, the results of this paper are convincing enough to support the basic objective and stance of this paper in its current version. Therefore, after a minor revision, this paper can be given a green signal to be published in journal 'Hydrology and Earth System Sciences (HESS)". Minor Comments: 1. The authors have used 25 GCMs in current study and all of them have different spatial resolution which has a lot of implications in results section. Therefore the authors are advised to explain how the spatial resolution of all those GCMs are made consistent with each other. 2. Page6 Line13: In current study, only high demand scenario has been chosen from a conservative perspective whereas the low demand scenarios has been discarded with the justification of declining rice-planting lands. However the authors did not provide any reference which supports author assumption of declining rice-planting lands. 3. Page6 Line11: Authors are suggested to please explain how they calculated economic growth and effectiveness so that it could be easy for readers to comprehend. 4. Page8 Line27: The line "The four free parameters of GR4J….inputs" is confusing and needs to be rephrased. Four free parameters has not been defined yet, therefore, to make it convenient for reader, please first define the four free parameters before abovementioned line. 5. Page14 Line14: Please rephrase "more water resources need be transferred" with "more water resources need to be transferred" 6. Page16 Line9: In line "More reliable risk estimates can be achieved from other uncertainty assessment methods though expensive efforts may be required" Please mention few other uncertainty assessment methods you are talking about so that it could be easy for readers to comprehend the context. 7. Page24 Figure 1: The annotation color in inset maps needs to be changed because its not clear enough.

---

## Referee Comment (RC2) · Anonymous Referee #2 · 31 Jul 2018

General Comments:

This manuscript describes a method of extending a bottom-up climate risk assessment by using logistic regression to estimate the probability that a water system will meet minimum performance criteria over a planning horizon based on the values of climate variables. The method is demonstrated through a case study of water management in the Geum River Basin in South Korea. The Geum River is host to two dams which are managed for water supply, flood control, and environmental flows. The case study analyzes two alternative operating policies' ability to meet both water supply goals and instream flow requirements under a broad range of potential changes in average

temperature, average precipitation, and precipitation variability. It is interesting to see the framework applied for multiple sub-basins within a larger system, and important to acknowledge uncertainty that an operating policy will meet a performance goal within specific climate scenarios.

The text is poorly written and organized, with many strangely used words that inhibit understanding. Key examples include "successive", "sub-component", and "risk of system failure," which are applied in ways that are not standard in the literature and never clearly defined. Many crucial details related to the methods and motivation do not become clear until carefully examining the results section. For example, I believed the logistic model was simply modeling the water supply/environmental flow reliability as a function of climate variables rather than the risk of falling short of the reliability threshold until carefully examining the figures and results. This was the main point of the manuscript, so it is critically important that it is immediately apparent upon reading the abstract and within every part of the manuscript. The text requires substantial rewording and re-organization to clearly summarize the methodological contribution and motivation earlier in the text, better define scientific notation, and ensure new words and concepts are defined clearly the first time they are introduced.

While the goal of the logistic model is a worthy one, it is not clear that the framework has been well executed in the case study or that the novel technical contribution bears sufficient relationship to the EEDS framework to be named for it. This lack of clarity may be a symptom of the confused text. However, based on my understanding of the case study, the methods used to execute the case study are flawed in several important ways. Further, the interpretation of results relies on questionable assumptions related to the fitness of GCM projections for water system risk assessment. Both the manuscript and analysis require major revisions.

Specific Comments:

Logistic regression model: (1) Limited calibration set: It is my understanding that the

logistic model was calibrated from 434 binary values that correspond to either water supply reliability or environmental flow reliability meeting a threshold under 434 unique combinations of three climate variables. If my understanding is correct, this would mean that there is one response (binary performance metric) per climate scenario (this should be clarified in the manuscript if that is incorrect). This is a very limited data set for analyzing risk of failure resulting from internal climate variability, especially given that each scenario-specific stochastic trace was (a) only 20 years long, and (b) initially identical to every other weather sequence in the analysis that had then been perturbed from the original trace to match a unique combination of average precipitation, average temperature, and precipitation coefficient of variation using quantile mapping. To characterize the effects of internal variability on risk of failure over a planning period, it would be preferable to use the binary reliability outcomes from many more stochastic realizations of weather sequences within each combination of climate variables. With a single stochastic trace perturbed into many climate scenarios, the modelled risk of failure is likely to be driven entirely by the climate scenario rather than the actual risk of missing a performance target under internal climate variability, and furthermore heavily biased across the climate response function by the single stochastic realization used to generate all climate scenarios. This seems to be the opposite of the intentions described in the introduction. (2) I do not see any part of the manuscript that assesses the performance of the logistic regression model using out-of-sample data. This is critical to the manuscript's success because it would provide evidence that the loss of information from modelling the risk of failing a satisficing criterion rather than evaluating the risk of failure through many simulations at each combination of climate variables could be worth the savings in computation time. (3) It is not clear whether there are separate logistic regression models for each sub-basin, performance metric, etc. How many logistic regression models are there in this case study? One per sub-basin, to model simultaneously meeting water supply reliability and environmental flow requirements? Two per sub-basin, each modelling risk of failing one of the objectives' minimum performance criterion? One, with sub-basins represented through dummy variables? If

the model is used to predict risk of failing mutual satisficing rather than risk of failing one performance threshold, would the model structure work if the two objectives were in tension (as in the Poff et al. 2015 case study) rather than aligned (as they are in this case study)? This section needs to clearly list the explanatory variables and document the dependent variables much more clearly.

Water system modelling framework: (1) Synthetic weather generator and streamflow temporal resolution: A daily weather generator is used to generate perturbed weather sequences and run them through a runoff model to generate streamflow. After simulating climate-changed streamflow using the runoff model, daily streamflows are aggregated to monthly flow. Why aggregate ex post rather than using a computationally cheaper weather generator and/or runoff model that is designed to operate at the monthly temporal resolution? (2) Temporal aggregation and precipitation coefficient of variation (cv): Perhaps the monthly streamflow resolution is the reason precipitation coefficient of variation was not a strong predictor of performance metrics? The authors should consider this possibility and potentially discard precipitation cv from their analysis, which might be better served by more stochastic realizations in each climate scenario rather than more climate variables. (3) Climate response surface: The sampling of average precipitation and precipitation coefficient of variation (cv) is coarse (20% increments). I suggest sampling these factors at tighter increments. (4) The computational expense of conducting bottom-up climate risk assessment is mentioned several times in the text. How computationally intense is the Geum water system model to evaluate?

Role of GCM projections in the case study: (1) GCMs are limited in their ability to simulate land/ocean/atmospheric mechanisms, especially those that take place at sub-grid scale resolution. This limits the information that can be credibly derived from projections for water resources planning. Precipitation coefficient of variation (CV), one of the climate variables used in the case study, is not well represented in GCMs so it is questionable to infer precipitation CV from GCM projections. This is why GCM projections

are not shown on some of the response surfaces in Poff et al. 2015 (in response to page 3, Line 18-19 of this manuscript). (2) This manuscript repeatedly mentions GCM counts as though GCM count in the feasible region on the climate response surface could be a decision criterion (e.g. page 3 line 19), and perhaps to some stakeholders it would be. However, this could also imply an attempt to quantify risk across the entire sampled climate space. Uncertainty quantification via ensembles of GCM projections is a challenging research question in its own right and would not be well treated by simply counting GCM projections from an arbitrary ensemble. Indeed, the point of bottom-up decision frameworks for climate risk management is avoiding this type of reliance on GCM projections with little scientific basis. Since this manuscript is designed to build on a bottom-up risk assessment framework, it is strange that so much emphasis is put on understanding performance under GCM projections in the text and figures.

Titling the framework: As mentioned above, it is not clear whether the logistic model is designed to model the risk of failing to mutually satisfice the eco-engineering performance thresholds or the risk of failing to meet one performance threshold. If the latter, the main technical contribution seems as appropriate for any single-objective climate response surface type risk assessment as for multi-objective climate response surface analyses, though it is applied here in a multi-objective climate response surface analysis. I would suggest the authors re-frame the analysis and revise the title to put the focus on the manuscript's main technical contribution, which is analyzing and communicating probabilistic information through a climate response surface (with an eco-engineering case study) rather than presenting a novel decision framework.

Technical corrections (typing errors, etc.)

Word choice: The meaning of the terms "successive", "risk of system failure", and "sub-components" in the context of this analysis is not clear from the text.

Page 2, line 31: Whatley et al. 2014 should be Whateley et al. 2014

Page 3, section 5: "However, all assessments using the response surfaces have focused on the "expected performance" rather than risk of system failure" Is this true? I thought many decision scaling papers evaluated reliability, which is risk of failure. . . Figure 9: Labels on X axis would be clearer in words. Also, isolating the results of the analysis to GCM projections is totally counter-intuitive here. The point of bottom-up climate response surface analyses is to avoid relying on GCMs in climate risk management. Figure 2: It is not clear where and how the logistic model comes into this framework based on Figure 2. Figures: None of the response surface figures include precipitation CV as one of the axes, though this is one of the sampled climate variables. The reasoning behind this should be clarified in the text.

---

## Author Comment (AC1) · 29 Aug 2018

We greatly appreciate valuable efforts of the referee 2. All the comments are sound and constructive, and we believe that they will improve our texts and assessments in the revision process. Specific responses are following as per comment.

General Comments: This manuscript describes a method of extending a bottom-up climate risk assessment by using logistic regression to estimate the probability that a water system will meet minimum performance criteria over a planning horizon based on the values of climate variables. The method is demonstrated through a case study of water management in the Geum River Basin in South Korea. The Geum River is host to

two dams which are managed for water supply, flood control, and environmental flows. The case study analyzes two alternative operating policies' ability to meet both water supply goals and instream flow requirements under a broad range of potential changes in average temperature, average precipitation, and precipitation variability. It is interesting to see the framework applied for multiple sub-basins within a larger system, and important to acknowledge uncertainty that an operating policy will meet a performance goal within specific climate scenarios. The text is poorly written and organized, with many strangely used words that inhibit understanding. Key examples include "successive", "sub-component", and "risk of system failure," which are applied in ways that are not standard in the literature and never clearly defined. Many crucial details related to the methods and motivation do not become clear until carefully examining the results section. For example, I believed the logistic model was simply modeling the water supply/environmental flow reliability as a function of climate variables rather than the risk of falling short of the reliability threshold until carefully examining the figures and results. This was the main point of the manuscript, so it is critically important that it is immediately apparent upon reading the abstract and within every part of the manuscript. The text requires substantial rewording and re-organization to clearly summarize the methodological contribution and motivation earlier in the text, better define scientific notation, and ensure new words and concepts are defined clearly the first time they are introduced. While the goal of the logistic model is a worthy one, it is not clear that the framework has been well executed in the case study or that the novel technical contribution bears sufficient relationship to the EEDS framework to be named for it. This lack of clarity may be a symptom of the confused text. However, based on my understanding of the case study, the methods used to execute the case study are flawed in several important ways. Further, the interpretation of results relies on questionable assumptions related to the fitness of GCM projections for water system risk assessment. Both the manuscript and analysis require major revisions.

–> The insufficient readability commented by the referee may be because we inadequately addressed the scientific meaning of the proposed combined framework in

the introduction and methodology sections. We will more clearly address the EEDS framework and potential challenges in stochastic uncertainty analyses. We will globally review the terminology used in our manuscript, and will improve its readability. Still, the contribution of our work is to combine the logistic regression with the bottom-up approach (not limited to the EEDS) so that users can efficiently quantify the risk of system failure. The value of the logistic regression in this work is to enable analyzers to efficiently gauge the probability of system failure without a large number of realizations for evaluating performance of a complex hydrologic system. We agree that internal climatic variability over a time horizon can significantly contribute to performance of hydrologic systems as shown in Whateley and Brown (2016). However, if we quantified the uncertainty from climatic variability using many (or long) stochastic generations, computational costs would not be small. When 100 random samplings are applied for a single climatic perturbation, computational costs increase by 100 times (in our case, it will become $343 \times 100$ tests). Although it is a simplified approach, the logistic regression makes it possible to gauge the risk of system failure in a collective manner using a single weather series for each climatic perturbation. This is a main contribution of our work. The bottom-right panel of Figure 4 shows the advantage of the logistic regression clearly. With -25% or larger changes in Pavg approximately, the river system is unlikely to satisfy the performance threshold (i.e., 100% of system failure). Between -25% and 10% changes in Pavg, the system can be either satisfactory or unsatisfactory (i.e., the risks of failure are between 100% and 0%). With +10% or greater changes in Pavg, the system seems to perfectly satisfy the threshold (0% of system failures). The number of failure (zero) decreases with increasing Pavg, while the number of success (one) increases. Importantly, variations of the stochastic weathers are different from one perturbation to another because of randomness given by the weather generator. In other words, even under different climatic variabilities across the 343 perturbations, it can be indicated that the risk of system failure declines with increasing Pavg. This can be modeled by the theoretical logistic equation that draws a smooth line between 100% failure (zero) and 100% success (one). We are not arguing this simplification

is perfect, but it can be efficient when quantifying the probability of system failures (or success). If one uses two explanatory variables (e.g., Pavg and Tavg), he or she can obtain the logistic surface of the probability of success such as the top-right panel of Figure 4 without additional stress tests. This approach is theoretically similar to the simple linear regression that approximates the variance of predicted values using the residuals from diverse data points. A single data point has only one residual and thus the variance of residuals cannot be obtained from it. However, a collection of residuals from many data points enables to quantify the variance of the residuals. Likewise, one outcome (i.e. 0 or 1) under a single climate perturbation does not allow us to quantify the risk of failure, but a collection of outcomes under various climate conditions enables it. We agree that many or long weather generations for one perturbation may quantify the risk of failure too, but it is not the only approach for assessing associated uncertainty. In revision, we can explain more clearly this contribution of our work. We will address some disadvantages and challenges in a typical stochastic uncertainty analysis and the bottom-up framework in the introduction. Then, we will describe the EEDS more clearly and how to incorporate the logistic regression into the framework in the methodology section. Thereafter, we will show the case study for Geum River Basin. This reorganization may improve readability of the manuscript. And, discussion and conclusions will be revised accordingly to highlight the scientific contribution of this work.

Specific Comments: Logistic regression model: (1) Limited calibration set: It is my understanding that the logistic model was calibrated from 434 binary values that correspond to either water supply reliability or environmental flow reliability meeting a threshold under 434 unique combinations of three climate variables. If my understanding is correct, this would mean that there is one response (binary performance metric) per climate scenario (this should be clarified in the manuscript if that is incorrect). This is a very limited data set for analyzing risk of failure resulting from internal climate variability, especially given that each scenario-specific stochastic trace was (a) only 20 years long, and (b) initially identical to every other weather sequence in the analysis

that had then been perturbed from the original trace to match a unique combination of average precipitation, average temperature, and precipitation coefficient of variation using quantile mapping. To characterize the effects of internal variability on risk of failure over a planning period, it would be preferable to use the binary reliability outcomes from many more stochastic realizations of weather sequences within each combination of climate variables. With a single stochastic trace perturbed into many climate scenarios, the modelled risk of failure is likely to be driven entirely by the climate scenario rather than the actual risk of missing a performance target under internal climate variability, and furthermore heavily biased across the climate response function by the single stochastic realization used to generate all climate scenarios. This seems to be the opposite of the intentions described in the introduction.

–> We agree that the internal variability significantly affect variation of the performance metrics. However, we disagree that multiple weather generations for each perturbation are required for quantifying the risk of failure. The theory of linear regressions unnecessarily requires many Y values for a single X value. Rather, many pairs of (X, Y) are needed to quantify the variance of residuals and the prediction intervals. Since the weather generator bootstraps the observed weathers for each perturbation, the 343 sets of the 20-year-long weather series contain different internal variability each other. In other words, the risk of failure can be obtained from diverse variability of the 343 weather series in a collective manner, not from a fixed one. It is true that long or many weather generations can quantify the internal variability of a single climatic perturbation rigorously. However, it is not the only approach for quantifying the risk of failure. Indeed, it can be time-consuming. We rather argue that there is no clear evidence to confirm that the risk estimates from the logistic regression were biased. We did not fix the variation of weather series for the 343 perturbations. The length of time horizon in this work (20 years) was determined following the definition of climate change given in the IPCC 4th assessment report. The IPCC defined it as the statistical changes during a decadal or a longer period, and thus we set the 20 years for generating weather series for each climatic perturbation. This length is not for capturing climatic variability

by random weather generations. Though one weather generation has a length of 20 years, the uncertainty from internal variability may be captured by collecting the 343 sets all together. In this work, the risk of system failure did not come from a single perturbation, but from integration of the 343 stress tests that contain different internal variabilities.

(2) I do not see any part of the manuscript that assesses the performance of the logistic regression model using out-of-sample data. This is critical to the manuscript's success because it would provide evidence that the loss of information from modelling the risk of failing a satisficing criterion rather than evaluating the risk of failure through many simulations at each combination of climate variables could be worth the savings in computation time.

–> We agree. It seems necessary to validate the risk estimate from the logistic regression. A possible approach is to compare one risk estimates at a selected perturbation from the logistic regression to the risk estimate from a number of random generations (e.g., 100 times) for the selected perturbation. This will add this validation in revision.

(3) It is not clear whether there are separate logistic regression models for each sub-basin, performance metric, etc. How many logistic regression models are there in this case study? One per sub-basin, to model simultaneously meeting water supply reliability and environmental flow requirements? Two per sub-basin, each modelling risk of failing one of the objectives' minimum performance criterion? One, with sub-basins represented through dummy variables? If the model is used to predict risk of failing mutual satisficing rather than risk of failing one performance threshold, would the model structure work if the two objectives were in tension (as in the Poff et al. 2015 case study) rather than aligned (as they are in this case study)? This section needs to clearly list the explanatory variables and document the dependent variables much more clearly.

–> Figure 5 shows the climatic bounds for 95% probability of success for each demand

node. So, the number of lines is same as the number of demand nodes. Figure 7 shows the climatic bounds for 95% of success for each instream flows location (seven in total). The logistic regression applied for each node and each instreamflow location. And, the highest bound for water supply and that for instreamflow were combined. In other words, if the most vulnerable demand node and the most vulnerable instreamflow location can have 95% probability of success under a certain climate stress, the other nodes and locations will have 95% or more probabilities automatically. The mutual zone made by the two highest bounds is the key information. We will explain more clearly in revision.

Water system modelling framework: (1) Synthetic weather generator and streamflow temporal resolution: A daily weather generator is used to generate perturbed weather sequences and run them through a runoff model to generate streamflow. After simulating climate-changed streamflow using the runoff model, daily streamflows are aggregated to monthly flow. Why aggregate ex post rather than using a computationally cheaper weather generator and/or runoff model that is designed to operate at the monthly temporal resolution?

–> This is due to validity of the hydrologic model. We needed a method for ungauged basins for each sub-basin, and already had a validated model. GR4J was validated by the LOOCV across South Korea by Kim et al. (2017). Though it is true that a monthly model is computationally efficient than daily models, another validation for ungauged basins will be required. Aggregating daily simulations was not very time-consuming, but the main computational cost in this work was the time required for 20-year-long sequential optimizations.

(2) Temporal aggregation and precipitation coefficient of variation (cv): Perhaps the monthly streamflow resolution is the reason precipitation coefficient of variation was not a strong predictor of performance metrics? The authors should consider this possibility and potentially discard precipitation cv from their analysis, which might be better served by more stochastic realizations in each climate scenario rather than more climate variables.

–> It is unlikely. Even with the temporal aggregation from daily to monthly values, the monthly flows were affected by Pcv. A higher Pcv resulted in larger streamflow, because precipitated water would reside in the soil for a shorter length due to more frequent high-intensity rainfall events, leading to less evapotranspiration. We found that Pcv was one of significant factors that explains the variation of total streamflow. However, it was not significant to explain the variation of the water supply reliability given by the 343 perturbations. We believe that the storage capacities of the sub-basin and the dams are likely factors that nullified the influence of Pcv on water availability. We will explain this more clearly in revision.

(3) Climate response surface: The sampling of average precipitation and precipitation coefficient of variation (cv) is coarse (20% increments). I suggest sampling these factors at tighter increments. (4) The computational expense of conducting bottom-up climate risk assessment is mentioned several times in the text. How computationally intense is the Geum water system model to evaluate?

–> Even with the interval we applied, changes in water supply performance seems to be sufficiently captured as shown in Figure 3a. However, in revision, the range and interval may be adjusted to zoom in the range between 500 and 1500 mm of Pavg (e.g., -60% to +60% at a 10% interval), because 1500+ mm of Pavg in Figure 3a mostly resulted in the maximum reliability (i.e., 1). We guess one week will be adequate to update the stress tests with the two scenarios, because all the models were readily available now. We believe that this would be possible during the revision process, and will improve this work.

Role of GCM projections in the case study: (1) GCMs are limited in their ability to simulate land/ocean/atmospheric mechanisms, especially those that take place at sub-grid scale resolution. This limits the information that can be credibly derived from projections for water resources planning. Precipitation coefficient of variation (CV), one of

the climate variables used in the case study, is not well represented in GCMs so it is questionable to infer precipitation CV from GCM projections. This is why GCM projections are not shown on some of the response surfaces in Poff et al. 2015 (in response to page 3, Line 18-19 of this manuscript).

–> We agree that the GCMs have limitations, and it is true that all the projections can be subject to significant uncertainty. It is not limited to Pcv. Pavg and Tavg may not be well captured by GCMs either. However, because of that reason, the bottom-up frameworks emerged by employing the stochastic tests imposing arbitrary climatic stresses on the hydrologic systems. And, overlaying GCMs projections on the response surfaces is a common approach to gauge climate change risks in most bottom-up assessments. Should we neglect Pcv values from GCMs because of the limitations in GCMs, even though the bottom-up framework was intended to consider the uncertainty in GCMs for practical decision makings (e.g., Brown et al., 2012)? We disagree that Pcv should not be overlaid on the response surfaces due to that uncertainty. Rather, we believe that Pcv values need to be overlaid as many as possible to combine the knowledge from the stress tests (i.e., response surface) and the climate sciences (i.e. GCMs). Without any reference points, the response surfaces can provide information of system sensitivity to climate stresses only. What if future climatic stresses are out of the range in which we can withstand? Poff et al. (2016) is a very innovative approach that allows quantifying multi-faceted system robustness to climate change. However, if any predictions are not combined with it, its usability may be limited in practice.

(2) This manuscript repeatedly mentions GCM counts as though GCM count in the feasible region on the climate response surface could be a decision criterion (e.g. page 3 line 19), and perhaps to some stakeholders it would be. However, this could also imply an attempt to quantify risk across the entire sampled climate space. Uncertainty quantification via ensembles of GCM projections is a challenging research question in its own right and would not be well treated by simply counting GCM projections from an arbitrary ensemble. Indeed, the point of bottom-up decision frameworks for climate risk

management is avoiding this type of reliance on GCM projections with little scientific basis. Since this manuscript is designed to build on a bottom-up risk assessment framework, it is strange that so much emphasis is put on understanding performance under GCM projections in the text and figures. Titling the framework: As mentioned above, it is not clear whether the logistic model is designed to model the risk of failing to mutually satisfice the eco-engineering performance thresholds or the risk of failing to meet one performance threshold. If the latter, the main technical contribution seems as appropriate for any single-objective climate response surface type risk assessment as for multi-objective climate response surface analyses, though it is applied here in a multi-objective climate response surface analysis. I would suggest the authors re-frame the analysis and revise the title to put the focus on the manuscript's main technical contribution, which is analyzing and communicating probabilistic information through a climate response surface (with an eco-engineering case study) rather than presenting a novel decision framework.

–> Perhaps, we put too much emphasis on the GCM counts in the text, though it seems to be intended by the original decision scaling framework (Brown et al., 2012). We will tone down in revision. Our point is that while the response surfaces of system performance is developed to consider risks (or uncertainty) associated with climate change, there is no quantified risk estimates in there interestingly. How do we get lessons from a response surface and climate projections? One of implications is where the locations of climate projections are on the response surface. It is natural for potential users to check if the projections are beyond the performance threshold or not. Thus, it was strange to us that there were no projections on the response surfaces in Poff et al. (2016). So, we guessed some GCMs projections might be located out of acceptable ranges in Poff et al. (2016). By its nature, a multi-purpose response surface should have a narrower acceptable range than a single-purpose one. For better indications from the multi-purpose response surfaces, we just suggest to convert them to the logistic surfaces directly indicating the risk of system failure and then overlay climate projections. In revision, we will better frame the manuscript and retitle it to improve

readability, as responded earlier.

Technical corrections (typing errors, etc.) Word choice: The meaning of the terms "successive", "risk of system failure", and "sub-components" in the context of this analysis is not clear from the text.

–> We will provide clear definitions for them.

Page 2, line 31: Whatley et al. 2014 should be Whateley et al. 2014

–> We will globally check mistypos.

Page 3, section 5: "However, all assessments using the response surfaces have focused on the "expected performance" rather than risk of system failure" Is this true? I thought many decision scaling papers evaluated reliability, which is risk of failure.

–> We will tone down. However, to our knowledge, many studies have usually developed the response surfaces in terms of the expected performance rather than the risk of failures even in the case that assessing uncertainty was a main objective (e.g., Kay et al., 2014).

Figure 9: Labels on X axis would be clearer in words. Also, isolating the results of the analysis to GCM projections is totally counter-intuitive here. The point of bottom-up climate response surface analyses is to avoid relying on GCMs in climate risk management.

–> We will describe the label more clearly. However, we disagree that the risk estimates from the GCMs are counter-intuitive. Then, how can we assess future climatic risks in practice? The response surface itself does not have predictions. If GCMs are discarded, only information from the response surfaces is system robustness. The probability of successive outcomes estimated from GCMs can be important information for decision-makers. Figure 9 shows the trade-off in future risks of system failures when changing the human-demand-only management policy.

Figure 2: It is not clear where and how the logistic model comes into this framework based on Figure 2. Figures: None of the response surface figures include precipitation CV as one of the axes, though this is one of the sampled climate variables. The reasoning behind this should be clarified in the text.

–> We will improve relevance of this figure to bring a better implication.

References

Brown, C., Ghile, Y., Laverty, M., and Li, K.: Decision scaling: linking bottom-up vulnerability analysis with climate projections in the water sector. Water Resour. Res., W09537, https://doi.org/10.1029/2011WR011212, 2012.

Kay, A. L., Crooks, S. M., and Reynard, N. S.: Using response surfaces to estimate impacts of climate change on flood peaks: assessment of uncertainty, Hydrol. Process., 28, 5273–5287, https://doi.org/ 10.1002/hyp.10000, 2014.

Poff, N. L., Brown, C. M., Grantham, T. E., Matthews, J. H., Palmer, M. A., Spence, C. M., Wilby, R. L., Haasnoot, M., Mendoza, G. F., Dominique, K. C., and Baeza, A.: Sustainable water management under future uncertainty with ecoengineering decision making, Nature Clim. Change, 6, 25-34. https://doi.org/10.1038/nclimate2765, 2016.

Whateley, S., and Brown C.: Assessing the relative effects of emissions, climate means, and variability on large water supply systems, Geophys. Res. Lett., 43, 11,329–11,338, http://doi.org/ 10.1002/2016GL070241, 2016.

---

## Author Comment (AC2) · 29 Aug 2018

We greatly appreciate the valuable comments from the referee 1, and will consider them to improve the manuscript in revision. Specific responses are following as per comments.

This paper has integrated the logistic regressions with the ecoengineering decision scaling framework to evaluate the risk of system failures in contrast to expected performance under dynamic climate change scenarios. This paper contains new insights and contains a lot of information for scientific community. However, the authors have explained the manuscript in complicated ways which I think could be explained in a

simplified manner therefore, the authors are advised to avoid using complex English sentences and try to make their next manuscript as simple as possible which would ultimately increase understanding as well as attract more readers. In a nutshell, the results of this paper are convincing enough to support the basic objective and stance of this paper in its current version. Therefore, after a minor revision, this paper can be given a green signal to be published in journal 'Hydrology and Earth System Sciences (HESS)".

–> By considering good suggestions from the referee 2, we will reorganize the manuscript by more clearly addressing how the EEDS can be challenged and how to improve it in the introduction and methodology sections. Then, we will then provide the case study for Geum River Basin. The terminology and the sentences used in the manuscript will be reviewed and improved. The scientific contribution of this work will be better emphasized by the revision. And, we will slightly refine the stress tests and adding validation of the risk estimate from the logistic regression.

Minor Comments: 1. The authors have used 25 GCMs in current study and all of them have different spatial resolution which has a lot of implications in results section. Therefore the authors are advised to explain how the spatial resolution of all those GCMs are made consistent with each other.

–> They were bias-corrected by a statistical downscaling method. We applied the de-trended quantile mapping (e.g., Eum and Cannon, 2017) as mentioned in P6L31. By the statistical downscaling, the climate hindcasts and forecasts were bias-corrected towards the observed climates during a reference period. Perhaps, the method is inadequately explained. We will add more explanation there.

2. Page6 Line13: In current study, only high demand scenario has been chosen from a conservative perspective whereas the low demand scenarios has been discarded with the justification of declining rice-planting lands. However the authors did not provide any reference which supports author assumption of declining rice-planting lands.

[Figure]

–> We will include some references to support decreasing rice-planting areas in South Korea (e.g., http://www.index.go.kr/potal/main/EachDtlPageDetail.do?idx_cd=1287). The decreasing trend is clearly indicated from the data. The reason why we selected the high demand scenario is to assess the worst scenario (i.e. drying climate plus high demand projections) as a case study.

3. Page6 Line11: Authors are suggested to please explain how they calculated economic growth and effectiveness so that it could be easy for readers to comprehend.

–> We will more clearly summarize the water plan 2020 for potential readers who have insufficient knowledge about the demand projection.

4. Page8 Line27: The line "The four free parameters of GR4J. . ..inputs" is confusing and needs to be rephrased. Four free parameters has not been defined yet, therefore, to make it convenient for reader, please first define the four free parameters before abovementioned line.

–> We will add one or two sentences to describe the model parameters.

5. Page14 Line14: Please rephrase "more water resources need be transferred" with "more water resources need to be transferred"

–> We will correct the sentence as advised.

6. Page16 Line9: In line "More reliable risk estimates can be achieved from other uncertainty assessment methods though expensive efforts may be required" Please mention few other uncertainty assessment methods you are talking about so that it could be easy for readers to comprehend the context.

–> This part will be revised by accepting a comment from the referee 2. We will validate the risk estimate from the logistic regression using a typical stochastic uncertainty analysis. So, this part will be revised accordingly.

7. Page24 Figure 1: The annotation color in inset maps needs to be changed because

its not clear enough.

–> We will improve readability of the figure 1.

---

## Author Response (AR1)

**Responses to Referee 1:**

We greatly appreciate the valuable comments, and they were very helpful to improve the manuscript. Specific responses are following as per comment.

1. This paper has integrated the logistic regressions with the ecoengineering decision scaling framework to evaluate the risk of system failures in contrast to expected performance under dynamic climate change scenarios. This paper contains new insights and contains a lot of information for scientific community. However, the authors have explained the manuscript in complicated ways which I think could be explained in a simplified manner therefore, the authors are advised to avoid using complex English sentences and try to make their next manuscript as simple as possible which would ultimately increase understanding as well as attract more readers. In a nutshell, the results of this paper are convincing enough to support the basic objective and stance of this paper in its current version. Therefore, after a minor revision, this paper can be given a green signal to be published in journal 'Hydrology and Earth System Sciences (HESS)'.

→ Considering constructive comments from Referee 2, we substantially revised the initial version of our manuscript. The manuscript is now retitled as "incorporating the logistic regression in a decision-centric framework for probabilistic assessment of climate change impacts on a complex water system". The revised manuscript was focused on the meaning of the logistic regression in decision-centric assessment.

Minor Comments:
1. The authors have used 25 GCMs in current study and all of them have different spatial resolution which has a lot of implications in results section. Therefore the authors are advised to explain how the spatial resolution of all those GCMs are made consistent with each other.
→ They were bias-corrected by a statistical downscaling method. We applied the detrended quantile mapping as described in P6L27-P7L6. By the statistical downscaling, the climate hindcasts and forecasts were bias-corrected towards the observed climates during a reference period. Perhaps, the method is inadequately explained. We will add more explanation there.

2. Page6 Line13: In current study, only high demand scenario has been chosen from a conservative perspective whereas the low demand scenarios has been discarded with the justification of declining rice-planting lands. However the authors did not provide any reference which supports author assumption of declining rice-planting lands.
3. Page6 Line11: Authors are suggested to please explain how they calculated economic growth and effectiveness so that it could be easy for readers to comprehend.
→ Comments 2 and 3 are all related to the demand data, but to improve conciseness of the manuscript, the parted that explained the demand projection was removed. This is because this information is available in the given reference (L6-17 in page 6). The revised manuscript is more focused on the logistic surfaces. The decreasing agricultural land is also presented in the given reference

4. Page8 Line27: The line "The four free parameters of GR4J. . ..inputs" is confusing and needs to be rephrased. Four free parameters has not been defined yet, therefore, to make it convenient for reader, please first define the four free parameters before abovementioned line.
→ In L8-14 in page 8, GR4J was re-described.

5. Page14 Line14: Please rephrase "more water resources need be transferred" with "more water resources need to be transferred"
→ In revision, the sentence was removed, and replaced with new discussion (from P12L22-P13L4).

6. Page16 Line9: In line "More reliable risk estimates can be achieved from other uncertainty assessment methods though expensive efforts may be required" Please mention few other uncertainty assessment methods you are talking about so that it could be easy for readers to comprehend the context.
→ In revision, we added a part for validation of the logistic regressions using the stochastic sampling (L28 in page 14 and Figure 9).

7. Page24 Figure 1: The annotation color in inset maps needs to be changed because its not clear enough.
→ This might be the resolution problem when converting the word document to a pdf format. We believe that the original figure file would be okay. If necessary, we will make a further improvement.

**Responses to Referee 2:**

We greatly appreciate valuable efforts of the referee 2. All the comments were sound and constructive to improve our manuscript in the revision process. Specific responses are following as per comment.

**General Comments:**
This manuscript describes a method of extending a bottom-up climate risk assessment by using logistic regression to estimate the probability that a water system will meet minimum performance criteria over a planning horizon based on the values of climate variables. The method is demonstrated through a case study of water management in the Geum River Basin in South Korea. The Geum River is host to two dams which are managed for water supply, flood control, and environmental flows. The case study analyzes two alternative operating policies' ability to meet both water supply goals and instream flow requirements under a broad range of potential changes in average temperature, average precipitation, and precipitation variability. It is interesting to see the framework applied for multiple sub-basins within a larger system, and important to acknowledge uncertainty that an operating policy will meet a performance goal within specific climate scenarios. The text is poorly written and organized, with many strangely used words that inhibit understanding. Key examples include "successive", "sub-component", and "risk of system failure," which are applied in ways that are not standard in the literature and never clearly defined. Many crucial details related to the methods and motivation do not become clear until carefully examining the results section. For example, I believed the logistic model was simply modeling the water supply/environmental flow reliability as a function of climate variables rather than the risk of falling short of the reliability threshold until carefully examining the figures and results. This was the main point of the manuscript, so it is critically important that it is immediately apparent upon reading the abstract and within every part of the manuscript. The text requires substantial rewording and re-organization to clearly summarize the methodological contribution and motivation earlier in the text, better define scientific notation, and ensure new words and concepts are defined clearly the first time they are introduced. While the goal of the logistic model is a worthy one, it is not clear that the framework has been well executed in the case study or that the novel technical contribution bears sufficient relationship to the EEDS framework to be named for it. This lack of clarity may be a symptom of the confused text. However, based on my understanding of the case study, the methods used to execute the case study are flawed in several important ways. Further, the interpretation of results relies on questionable assumptions related to the fitness of GCM projections for water system risk assessment. Both the manuscript and analysis require major revisions.

→ We substantially revised the manuscript to present the main point from the case study. The revised manuscript retitled as "Incorporating the logstic regression into a decision-centric framework for probabilistic assessment of climate change impacts on a complex water system". This may be better than using the term "logistic EEDS". The revised manuscript focuses on assessing the probabilities of system failures (i.e. system performances less than pre-defined thresholds) using the logistic regressions. The varying performances across the river basin is now not a major part of this work. The terms "successive" and "sub-components" were not used in the revision, while the risk of system failures was still in use. We believe it can be viewed as the probability of unsatisfactory performances against the threshold. Now, the EEDS framework is a case study for an application of the proposed approach.
To improve the manuscript, we revised the introduction to emphasize the uncertainty associated with the response surfaces of performance metrics with several prior studies. This implies that risks of unsatisfactory performances can exist even within the climate zone of satisfactory performance in the response surfaces. Incorporating the logistic regression may be an approximate approach to supplement this weakness of the response surfaces. For validating the logistic regressions, we conducted stochastic analysis using the same system models but with arbitrary climate perturbations that were not included in the logistic regressions (Figure 9). The probabilities of satisfactory performances from the stochastic samplings and the logistic model approximately agreed. This will improve the validity of the proposed approach.
The contribution of our work is to combine the logistic regression with the bottom-up approach (not limited to the EEDS) so that users can efficiently quantify the probability of system failures. The value of the logistic regression is to enable to estimates the probability without a large number of realizations. If one estimate it using many (or long) stochastic generations, computational costs would be expensive. When 100 random samplings are applied for a single climatic perturbation, computational costs increase by 100 times. The logistic regression can help to solve this problem. Although it is an approximate approach, the internal variation of 20-year-long stochastic weathers is different from one perturbation to another used in the stress tests. Thus, the impacts of internal climate variation

would be captured in a collective manner by the logistic regression. In revision, we used 539 perturbations for the stress tests with a shorter interval for the mean precipitation. It will reduce the concern that the referee commented.

**Specific Comments:**
Logistic regression model: (1) Limited calibration set: It is my understanding that the logistic model was calibrated from 434 binary values that correspond to either water supply reliability or environmental flow reliability meeting a threshold under 434 unique combinations of three climate variables. If my understanding is correct, this would mean that there is one response (binary performance metric) per climate scenario (this should be clarified in the manuscript if that is incorrect). This is a very limited data set for analyzing risk of failure resulting from internal climate variability, especially given that each scenario-specific stochastic trace was (a) only 20 years long, and (b) initially identical to every other weather sequence in the analysis that had then been perturbed from the original trace to match a unique combination of average precipitation, average temperature, and precipitation coefficient of variation using quantile mapping. To characterize the effects of internal variability on risk of failure over a planning period, it would be preferable to use the binary reliability outcomes from many more stochastic realizations of weather sequences within each combination of climate variables. With a single stochastic trace perturbed into many climate scenarios, the modelled risk of failure is likely to be driven entirely by the climate scenario rather than the actual risk of missing a performance target under internal climate variability, and furthermore heavily biased across the climate response function by the single stochastic realization used to generate all climate scenarios. This seems to be the opposite of the intentions described in the introduction.
→ As replied to the general comment, we added a validation of the logistic model by testing 100 weather generations for three arbitrarily chosen perturbations that were not used for logistic model. Figure 9 shows that the probability estimates from the logistic model and the stochastic sampling approximately agreed. We believe this would improve validity of the logistic surfaces. Even though one set of weather series was generated for a single climate perturbation, 539 perturbations should have different internal variability due to randomness. Thus, the logistic regression can capture the impact of internal climatic variability in a collective manner.

(2) I do not see any part of the manuscript that assesses the performance of the logistic regression model using out-of-sample data. This is critical to the manuscript's success because it would provide evidence that the loss of information from modelling the risk of failing a satisficing criterion rather than evaluating the risk of failure through many simulations at each combination of climate variables could be worth the savings in computation time.
→ All the logistic regressions applied to the case study are summarized in Table A2 with the McFadden R2. As replied to comment 1, 3 sets of out of sample data were used for validation.

(3) It is not clear whether there are separate logistic regression models for each subbasin, performance metric, etc. How many logistic regression models are there in this case study? One per sub-basin, to model simultaneously meeting water supply reliability and environmental flow requirements? Two per sub-basin, each modelling risk of failing one of the objectives' minimum performance criterion? One, with subbasins represented through dummy variables? If the model is used to predict risk of failing mutual satisficing rather than risk of failing one performance threshold, would the model structure work if the two objectives were in tension (as in the Poff et al. 2015 case study) rather than aligned (as they are in this case study)? This section needs to clearly list the explanatory variables and document the dependent variables much more clearly.
→ All the logistic regressions applied to the case study are presented in Table A2. This will prevent the confusion.

Water system modelling framework:
(1) Synthetic weather generator and streamflow temporal resolution: A daily weather generator is used to generate perturbed weather sequences and run them through a runoff model to generate streamflow. After simulating climate-changed streamflow using the runoff model, daily streamflows are aggregated to monthly flow. Why aggregate ex post rather than using a computationally cheaper weather generator and/or runoff model that is designed to operate at the monthly temporal resolution?
→ This was because validity of the hydrological model. We needed a method for ungauged basins for each sub-basin, and already had a validated model. GR4J was validated by the LOOCV across South Korea by Kim et al. (2017). Though it is true that a monthly model is computationally efficient than daily models, another validation for ungauged basins will be required. Aggregating daily simulations was not very time-consuming, but the main

computational cost in this work was the time required for 20-year-long sequential optimizations. We did not change the model for revision.

(2) Temporal aggregation and precipitation coefficient of variation (cv): Perhaps the monthly streamflow resolution is the reason precipitation coefficient of variation was not a strong predictor of performance metrics? The authors should consider this possibility and potentially discard precipitation cv from their analysis, which might be better served by more stochastic realizations in each climate scenario rather than more cli mate variables.
→ It is unlikely. Even with the temporal aggregation from daily to monthly values, the monthly flows were affected by $P_{cv}$. A higher $P_{cv}$ resulted in larger streamflow, because precipitated water would reside in the soil for a shorter length due to more frequent high-intensity rainfall events, leading to less evapotranspiration. We found that $P_{cv}$ was one of significant factors that explain the variation of total streamflow. However, it was not significant to explain the variation of the water supply reliability to 539 perturbations. It is explained in L18-28 in page 10.

(3) Climate response surface: The sampling of average precipitation and precipitation coefficient of variation (cv) is coarse (20% increments). I suggest sampling these factors at tighter increments. (4) The computational expense of conducting bottom-up climate risk assessment is mentioned several times in the text. How computationally intense is the Geum water system model to evaluate?
→ In revision, the perturbations for $P_{avg}$ were adjusted to (e.g., -60% to +40% at a 10% interval). While we reached the same conclusions, this refinement would improve reliability of the logistic surfaces.

Role of GCM projections in the case study:
(1) GCMs are limited in their ability to simulate land/ocean/atmospheric mechanisms, especially those that take place at sub-grid scale resolution. This limits the information that can be credibly derived from projections for water resources planning. Precipitation coefficient of variation (CV), one of the climate variables used in the case study, is not well represented in GCMs so it is questionable to infer precipitation CV from GCM projections. This is why GCM projections are not shown on some of the response surfaces in Poff et al. 2015 (in response to page 3, Line 18-19 of this manuscript).
→ It is true that GCM projections are subject to significant uncertainty. It is not limited to Pcv. $P_{avg}$ and $T_{avg}$ may not be well captured by GCMs either. However, because of that reason, the bottom-up frameworks emerged by employing the stochastic tests imposing arbitrary climatic stresses on the hydrologic systems. Overlaying GCMs projections on the response surfaces is a common approach to gauge climate change risks in the bottom-up assessments. Superimposing GCMs on the logistic surfaces can provide information of future climate risks together with system robustness to climate change. It is discussed in L28 in page 14.

(2) This manuscript repeatedly mentions GCM counts as though GCM count in the feasible region on the climate response surface could be a decision criterion (e.g. page 3 line 19), and perhaps to some stakeholders it would be. However, this could also imply an attempt to quantify risk across the entire sampled climate space. Uncertainty quantification via ensembles of GCM projections is a challenging research question in its own right and would not be well treated by simply counting GCM projections from an arbitrary ensemble. Indeed, the point of bottom-up decision frameworks for climate risk management is avoiding this type of reliance on GCM projections with little scientific basis. Since this manuscript is designed to build on a bottom-up risk assessment framework, it is strange that so much emphasis is put on understanding performance under GCM projections in the text and figures. Titling the framework: As mentioned above, it is not clear whether the logistic model is designed to model the risk of failing to mutually satisfice the eco-engineering performance thresholds or the risk of failing to meet one performance threshold. If the latter, the main technical contribution seems as appropriate for any single-objective climate response surface type risk assessment as for multi-objective climate response surface analyses, though it is applied here in a multi-objective climate response surface analysis. I would suggest the authors re-frame the analysis and revise the title to put the focus on the manuscript's main technical contribution, which is analyzing and communicating probabilistic information through a climate response surface (with an eco-engineering case study) rather than presenting a novel decision framework.
→ Considering this comment, the manuscript was retitled. By its nature, a multi-purpose response surface should have a narrower acceptable range than a single-purpose one. For better indications from the multi-purpose response surfaces, we just attempted to convert them to the logistic surfaces directly indicating the risk of system failure and then overlay climate projections. This point is highlighted in the revised manuscript, rather than focusing on the GCM counts.

Technical corrections (typing errors, etc.)
Word choice: The meaning of the terms "successive", "risk of system failure", and "sub-components" in the context of this analysis is not clear from the text.
→ As replied, in the revised manuscript, we did not use the terms "successive" and "sub-components", but "risk of system failure" is still in use. It can be viewed as the probability of unsatisfactory performances throughout the manuscript.

Page 2, line 31: Whatley et al. 2014 should be Whateley et al. 2014 –> We will globally check mistypos.
→ We globally checked the mistypos, and re-listed the references.

Page 3, section 5: "However, all assessments using the response surfaces have focused on the "expected performance" rather than risk of system failure" Is this true? I thought many decision scaling papers evaluated reliability, which is risk of failure.
→ Instead, we emphasized the uncertainty in the response surfaces in the introduction (L26 in page 2).

Figure 9: Labels on X axis would be clearer in words. Also, isolating the results of the analysis to GCM projections is totally counter-intuitive here. The point of bottom-up climate response surface analyses is to avoid relying on GCMs in climate risk management.
→ Figure 9 was removed in the revision. Instead, we explained the trade-off between the location E and the sub-basin 2 when considering the instreamflow requirement using the logistic regressions in Figure 7.

Figure 2: It is not clear where and how the logistic model comes into this framework based on Figure 2. Figures: None of the response surface figures include precipitation CV as one of the axes, though this is one of the sampled climate variables. The reasoning behind this should be clarified in the text.
→ Now, the EEDC framework is narratively described rather than using Figure.

**Incorporating the logistic regression into a decision-centric framework for probabilistic assessment of climate change impacts on a complex water system**

Daeha Kim[1], Jong Ahn Chun[1], Si-Jung Choi[2]

[1]APEC Climate Center, Busan, 48058, South Korea
[2]Korea Institute of Civil Engineering and Building Technology, Gyeonggi-do, 10223, South Korea

*Correspondence to*: Si-Jung Choi (sjchoi@kict.re.kr)

~~**Abstract.** Climate change is a global stressor that can undermine water management policies developed with the assumption of stationary climate, necessitating robust strategies for operating water infrastructures under uncertain climate and socioeconomic conditions. In this study, we modified the eco-decision scaling framework to assess water supply and environmental reliabilities varying across a large river basin in response to changing climate exposures. The ordinary climate responses functions of the decision scaling framework were replaced with the logistic response surfaces against pre-defined thresholds to measure risks of non-successive outcomes indicated by individual climate projections. The logistic response surfaces of water supply performances and those of environmental reliabilities at sub-systems within the river basin were all combined to gauge potential risks of climate change from multiple perspectives. The case study for a large river basin in South Korea showed water supply reliabilities at the sub-basins were expected to be satisfactory against the water demand projections to 2030, while the human-demand-only operations could lower the environmental reliabilities for 2020-2039. To reduce the environmental risks, the stakeholders should accept increasing risks of unsatisfactory water supply at the sub-basins with small water demands. This study highlights that the logistic eco-engineering decision scaling would better support risk-based decision making processes from multiple perspectives by allowing users to measure potential risks of unsatisfactory outcomes together with visualization of various system responses to climatic stresses in a complex hydrologic system.~~

**Abstract.** Climate change is a global stressor that can undermine water management policies developed with the assumption of stationary climate. While the response-surface-based assessments provided a new paradigm for formulating actionable adaptive solutions, the uncertainty associated with the stress tests pose challenges. To address the risks of unsatisfactory performances in a domain of climate stresses, this study proposed to incorporate the logistic regression into a decision-centric framework. The proposed approach replaces the "responses surfaces" of the performance metrics typically used for the decision scaling framework with the "logistic surfaces" that describes the risk of system failures against pre-defined

[revised manuscript text omitted]

Nonetheless, the bottom-up approaches have several shortcomings that may mislead to regretful outcomes, since climate predictions are not the only uncertainty source in a decision-making process. Steinschneider et al. (2015b) argued that uncertainty stemming from hydrologic modelling (e.g., rainfall-runoff modelling and system optimizations) commonly used in the stress tests could be comparable to those from climate predictions. Kay et al. (2014) also emphasized the necessity of uncertainty allowances to be used alongside the response surfaces of flood risk indicators. These studies suggest that simplifications and assumptions necessary for the stress tests, which underpin the bottom-up assessment, could misguide decision-makers to inappropriate and/or untimely strategies, necessitating additional risk information of non-successive outcomes together with performance metrics projected by GCMs.

Furthermore, one may cast a concern that the response-surface-based approachesAmong the decision-centric frameworks, the assessments based on the response functions of system performance have provided convenience to define decision thresholds at which adaptation actions are required (e.g., Kim et al., 2018; Steinschneider et al., 2015a; Turner et al., 2014; Whateley et al., 2014; Brown et al., 2012; Prudhomme et al., 2010)Prudhomme et al., 2010; Brown et al., 2012; Steinschneider et al., 2015a; Whateley et al., 2014). They developed the relationships between system performances and climatic stressors (hereafter referred to as the "response surfaces") via stress tests. Then, GCM projections were employed to indicate future system performance on the response surfaces. The response-surface-based methods have been refined to consider spatially varying system performance (e.g., Schlef et al., 2018) and multiple management objectives within a

hydrological system (e.g., Cully et al., 2016). They allowed efficient evaluation of climate change risks simply by comparing the performance metrics indicated by a collection of GCMs against pre-defined thresholds.

Nonetheless, uncertainty of the response surfaces cannot be neglected due to assumptions and simplifications associated with the stress tests (Kay et al., 2014). Indeed, Steinschneider et al. (2015b) argued that hydrological modelling and climatic variability may introduce uncertainty in the response surfaces as much as GCM projections. Kim et al. (2018) showed how climate change risks can be underestimated when a modelling scale was inappropriately chosen. Hence, over-reliance on the response surfaces of general performance metrics may misguide users to inappropriate and/or untimely adaptation policies. Importantly, the response surfaces have usually been developed with climatic shifts defined by long-term changes in statistical moments of weather observations (e.g., Kim et al., 2018; Poff et al., 2016; Steinschneider et al., 2015a; Whateley et al., 2014), even though they might insufficiently explain variation of chosen performance indicators. Whateley and Brown (2016) found that water supply system performance can be attributed mostly to uncertainty in internal climate variability over a time horizon of policy planning.

The prior studies imply that risks of system failures still exists even in the climate zone of satisfactory performance in the response surfaces. This uncertainty issue may be mended in part by evaluating the risks of system failures along with the response surfaces of expected performances. While it is possible to conduct stochastic uncertainty analyses with the stress tests (e.g., Steinschneider et al., 2015b; Whateley and Brown, 2016), this approach would require expensive computational costs even with modern computing power (Whateley et al., 2016). In this work, therefore, an efficient approach was proposed to evaluate the risks of system failures within a decision-centric framework. We simply incorporated the logistic regression into typical stress tests for the response-surface-based assessments. As a case study, here we provided a slightly modified version of the eco-engineering decision scaling framework (Poff et al., 2016) to explore the probabilities of system failures varying across a complex river system with two contrasting management purposes.

may over rely on the chosen performance indicator. If it is developed with a general performance metric of hydrologic systems (e.g., Turner et al., 2014), the response surface may not conceive failures at their sub-systems or those from other standpoints hidden behind successive general performance. The performance of water resources systems can vary across scales (e.g., entire system vs. sub-systems) and standpoints (e.g., for human demands vs. for environmental requirements), since practical management options are beyond the optimality of general performance due to uncertain hydrologic and socioeconomic conditions (Herman et al., 2014; Rosenberg, 2015). To overcome this weakness, Steinschneider et al. (2015a) expanded the decision scaling framework by developing multiple response surfaces of various indicators. Cully et al. (2016) embedded the multi-objective failure boundaries in the response surfaces of operational performances in a reservoir system. Schlef et al. (2017) investigated spatio-temporal variability of sub-components' vulnerability in a complex regulated river system.

A limitation of the abovementioned studies, however, is that users may face significantly increased dimensionality when many scales and standpoints are considered. If many adaptation options are preselected for the stress tests, increasing dimensionality of the response surfaces would become far more problematic. To this dimensionality problem, the ecodecision scaling framework (Poff et al., 2016) may be a solution. Poff et al. (2016) proposed to overlap multiple response surfaces representing engineering and ecological performances so as to identify climate zones within which all the performance criteria are mutually satisfied. This approach enables users to gauge the climatic tolerance of given systems from multiple perspectives. Likewise, one can evaluate system performances varying across scales and standpoints by integrating their response surfaces into one single climatic domain. Despite simplicity and usefulness of the eco-decision scaling, few studies have taken this flexible framework for assessing performance of water systems under non-stationary climate.

In this work, therefore, we attempted to assess impacts of climate change on water supply capacity of a regulated river basin using the eco-engineering decision scaling framework in combination with the logistic regression analysis. Here, performance indicators varying across the complex river basin were evaluated using multiple logistic response surfaces to provide probabilities of successive outcomes in response to climate stressors. For the bottom-up assessment, we embedded a sequential optimization scheme in stochastic stress tests to quantify water supply reliabilities under diverse climate stresses. The performances of the entire system and its sub-components gained from the stress tests were visualized all together in a single climatic space for better understanding of the subsystem's responsive behaviours to climatic changes. Numerous GCM projections were then superimposed on the combined response surface to evaluate future system performances under projected climate scenarios.

[revised manuscript text omitted]

In this study, we slightly modified this framework by converting the response surfaces of performance metrics using the logistic regressions between binary outcomes against the threshold and corresponding climatic perturbations. This conversion allows users to estimate 
[revised manuscript text omitted]

[Figure]

[Figure]

**Figure 76: As in Figure 54, but for  $\rho_c > 0.70 \pi_{c70} = 95\%$ at the seven locations .**

[Figure]

Figure 7: The logistic regression models for $\pi_{s95}$ at the sub-basin 3002 (left) and $\pi_{e70}$ at the location E (right) under the human-demand-only operation (upper) and the operation considering human demands and instreamflow together (lower)

[Figure]

[Figure]

**Figure 88: Climatic bounds for $\pi_{s95}$ = 95% (left) and $\pi_{e70}$ = 95% (middle), and the bound mutually satisfying both criteria (right) under the human-demands-only operations (top) in comparison to the operations considering the instream flows at the location E (bottom). The empty and filled circles are the 50 GCM projections for 2020-2039. The symbols on the right panels are the 50 GCM projections and the arrows indicate climate change to the ensemble of the GCMs for 2020-2039.**

[Figure]

**Figure 9:** 1:1 plots between the probabily estimates from the stochastic sampling and those from the logistic regressions for the case study.Box plots of $\pi_{e,95}$ and $\pi_{e,70}$ estimated from the 50 GCM projections for 2020-2039 (a) with the human-demands-only operations and (b) with the operations considering the instream flow requirement at the location E.

---

## Referee Report (RR1)

Page 1, Lines 15-20: I think the "should" in "stakeholders should accept the risk…" should be changed to "could." This implies a possible choice in a more neutral way.

Page 2, line 5: Do the authors really mean "underutilization" here? This implies that there are good reasons to use GCM projections more, but practitioners choose not to use them despite this. I suggest rephrasing to "less utilization of GCM-led strategies" unless I have misunderstood.

Introduction is much clearer and makes sense. I'm still not sold on whether logistic regression is actually makes sense given their argument that a stochastic metric would be too computationally expensive to evaluate across the response surface.

Page 6: Is "instreamflows" the correct spelling here? Perhaps "instream flows"?

Page 6, line 14-17: Could also be interesting to perturb demand across a wider range of uncertainty, but I understand there are limits.

Page 7, line 15 - Page 8, line 6: So only one stochastic series was really generated, then it was perturbed in 539 flavors of precipitation CV, mean, and mean temperature? The explanation of the logistic model on Page 8 lines 4-6 is jarringly brief- perhaps mention that the model is described in more detail further on. This section also needs clearer explanation of exactly what the weather generator simulations were here (i.e., there was only one sampled time series) for those who are not already familiar with it.

Page 7, line 29-Page 8, line 6: Please say from the beginning how long each simulated weather series is. Based on Page 9, it looks like each weather series is 20 years long. The phrasing on page 7- "three bidecadal properties"- is confusing. The reader may interpret that to mean that the time series is longer than a decade, and each of the properties is calculated twice a decade- or the series is even longer and the property is calculated every 20 years, or for 20 random years within a longer time series. Please state clearly that the stochastic simulation was 20 years long, and that the properties are calculated based on the entire perturbed series.

Page 7, line 29-Page 8, line 6: The range of climate change evaluated here is VERY wide, to the point that it might be preferable to refine the change increments (e.g. 5% change instead of 10% change) to get a better idea of the response surface.

Page 8, line 29: Please change "539 stochastic weather sets" to "539 climate-altered versions of the stochastic weather simulation" if that accurately describes the set.

Page 9, line 30: "20-year" is much clearer than "bidecadal" (Page 7)

Page 14, lines 1-19: This strengthens the paper, but belongs in the "Results" section with more detail about the analysis. For example, were there really 300 stochastic simulations, or 100 stochastic simulations that were perturbed to match each of the three climate perturbation examples? What was the range of deviation between the logistic model's prediction and risk evaluated through the 100 simulations? Did the logistic model's skill vary substantially among the three climate perturbation examples? This analysis is what convinces the reader that the

paper's main technical contribution could be worthwhile, so I suggest the authors elaborate on the results of this analysis as suggested above.

I also suggest that the authors replace this space with a discussion of the analysis. What are the implications of the logistic model's predictive skill relative to that evaluated through the stochastic simulations for water management? What would a water manager think about this- when would a computationally efficient logistic model be worthwhile despite the less rigorous exploration of internal climate variability? What would the authors say to a water manager about applying the results of this analysis to the water manager's work? How would the authors advise other practitioners applying this logistic model- would they suggest a similar validation of the logistic results against an in-depth exploration of a few scenarios?

Figure 9: There were 100 simulations at three locations- shouldn't there be 300 points on this plot? It does not look like there are. Please add more information to the caption- the readers should understand what they are being shown here. The caption should include the three climate perturbations the points come from and list the number of points.

---

## Author Response (AR2)

**Responses to Referee 2:**

We greatly thank for the valuable efforts of Dr. Caitlin Spence to improve our manuscript. The constructive comments led us to carefully review our results. For validation of the logistic models, we used another internal scheme that reuses the resamples for regression analyses. This would improve the validity of the models in this study. We generally accepted Dr. Spence's suggestions in the second round of revision process. Specific responses are following as per comment.

Page 1, Lines 15-20: I think the "should" in "stakeholders should accept the risk…" should be changed to "could." This implies a possible choice in a more neutral way.
→ We revised it as suggested (P1L16).

Page 2, line 5: Do the authors really mean "underutilization" here? This implies that there are good reasons to use GCM projections more, but practitioners choose not to use them despite this. I suggest rephrasing to "less utilization of GCM-led strategies" unless I have misunderstood.
→ We revised it as "hindering utilization of the GCM-led strategies" (P2L6).

Introduction is much clearer and makes sense. I'm still not sold on whether logistic regression is actually makes sense given their argument that a stochastic metric would be too computationally expensive to evaluate across the response surface.
→ We exemplified the Monte-Carlo approach to gauge the uncertainty of response surfaces. When applying the typical stochastic uncertainty analysis for the stress test, the computational costs could be multiplied manyfold (P2L5-L11).

Page 6: Is "instreamflows" the correct spelling here? Perhaps "instream flows"?
→ We globally corrected it throughout the manuscript.

Page 6, line 14-17: Could also be interesting to perturb demand across a wider range of uncertainty, but I understand there are limits.
→ We understand the necessity of perturbing the water demand, because socioeconomic conditions could change in various directions. However, it is beyond the scope of this work, and may require a different framework for assessments. We will consider this valuable comment in further studies.

Page 7, line 15 - Page 8, line 6: So only one stochastic series was really generated, then it was perturbed in 539 flavors of precipitation CV, mean, and mean temperature? The explanation of the logistic model on Page 8 lines 4-6 is jarringly brief- perhaps mention that the model is described in more detail further on. This section also needs clearer explanation of exactly what the weather generator simulations were here (i.e., there was only one sampled time series) for those who are not already familiar with it.

→ We explained more clearly about the weather generations (P8L6-L15). It is emphasized that **all the stochastic weather series are of different internal variability,** and the 539 climatic perturbations were more clearly explained. Yet, we did not include more descriptions of the WG, because they are available in the given references. The core components of the WG (i.e., Wavelet AR, Markov chain, and Quantile Mapping) are summarized in P7L26-P8L5. I think too much detail on this tool would rather distract potential readers to catch the focus of this work.

Page 7, line 29-Page 8, line 6: Please say from the beginning how long each simulated weather series is. Based on Page 9, it looks like each weather series is 20 years long. The phrasing on page 7- "three bidecadal properties"- is confusing. The reader may interpret that to mean that the time series is longer than a decade, and each of the properties is calculated twice a decade- or the series is even longer and the property is calculated every 20 years, or for 20 random years within a longer time series. Please state clearly that the stochastic simulation was 20 years long, and that the properties are calculated based on the entire perturbed series.
→ We clearly indicated it as "the 20-year-long precipitation and temperature series" in P8L6. Whenever necessary, we rephrased "bi-decadal" to "20 years" across the manuscript. We are now saying, "…, each weather series were represented with the mean annual precipitation ($P_{avg}$), the CV of daily precipitation ($P_{cv}$), and the mean annual temperature ($T_{avg}$) over the 20-year time horizon." (P8L14). This may be clearer.

Page 7, line 29-Page 8, line 6: The range of climate change evaluated here is VERY wide, to the point that it might be preferable to refine the change increments (e.g. 5% change instead of 10% change) to get a better idea of the response surface.
→ Seemingly, the interval appears to be wide. However, as shown in Figure 3, it was sufficiently narrow to develop the regression models. There is a trade-off in narrowing the interval. If we chose a smaller interval, the computational costs would become more burdensome for the stress test. We think it was efficient to build the logistic models.

Page 8, line 29: Please change "539 stochastic weather sets" to "539 climate-altered versions of the stochastic weather simulation" if that accurately describes the set.
→We corrected the sentence as suggested (P9L5).

Page 9, line 30: "20-year" is much clearer than "bidecadal" (Page 7)
→ As relied, we corrected "bi-decadal" throughout the manuscript.

Page 14, lines 1-19: This strengthens the paper, but belongs in the "Results" section with more detail about the analysis. For example, were there really 300 stochastic simulations, or 100 stochastic simulations that were perturbed to match each of the three climate perturbation examples? What was the range of deviation between the logistic model's prediction and risk evaluated through the 100 simulations? Did the logistic model's skill vary substantially among

the three climate perturbation examples? This analysis is what convinces the reader that the paper's main technical contribution could be worthwhile, so I suggest the authors elaborate on the results of this analysis as suggested above.

I also suggest that the authors replace this space with a discussion of the analysis. What are the implications of the logistic model's predictive skill relative to that evaluated through the stochastic simulations for water management? What would a water manager think about this- when would a computationally efficient logistic model be worthwhile despite the less rigorous exploration of internal climate variability? What would the authors say to a water manager about applying the results of this analysis to the water manager's work? How would the authors advise other practitioners applying this logistic model- would they suggest a similar validation of the logistic results against an in-depth exploration of a few scenarios?

→We moved the validation to the result section (P13L15-L31). Although the validation was from only three chosen stresses, they are out of the samples used in the logistic models. So, it could be regarded as an "external validation". The median and the highest differences between the two estimates were 0.004 and 0.15, respectively. However, due to expensive computational costs, it was difficult to increase the outside samples. The 300 Monte Carlo simulations are already burdensome when comparing to the 539 samples used for the logistic models. Hence, we added an internal validation (the bootstrap prognoses), providing an indication that the logistic models were of acceptable predictive performance. The validations were discussed in the section 5.1. As advised, we addressed advantages of the logistic models in practice too.

Figure 9: There were 100 simulations at three locations- shouldn't there be 300 points on this plot? It does not look like there are. Please add more information to the caption- the readers should understand what they are being shown here. The caption should include the three climate perturbations the points come from and list the number of points.

→ Since we compared the probabilities of success at the sub-basins and the instream flow locations, the total number of points should be 3 perturbations * (14 sub-basins + 7 instream flow locations) = 63. We added this in the caption of the Figure 8 as advised.